# Tackling GNARLy Problems: Graph Neural Algorithmic Reasoning Reimagined through Reinforcement Learning

## Abstract

Neural Algorithmic Reasoning (NAR) is a paradigm that trains neural networks to execute classic algorithms by supervised learning. Despite its successes, important limitations remain: inability to construct valid solutions without post-processing and to reason about multiple correct ones, poor performance on combinatorial NP-hard problems, and inapplicability to problems for which strong algorithms are not yet known. To address these limitations, we reframe the problem of learning algorithm trajectories as a Markov Decision Process, which imposes structure on the solution construction procedure and unlocks the powerful tools of imitation and reinforcement learning (RL). We propose the GNARL framework, encompassing the methodology to translate problem formulations from NAR to RL and a learning architecture suitable for a wide range of graph-based problems. We achieve very high graph accuracy results on several CLRS-30 problems, performance matching or exceeding much narrower NAR approaches for NP-hard problems and, remarkably, applicability even when lacking an expert algorithm.

## 1 Introduction

Neural Algorithmic Reasoning (NAR) is a framework that aims to achieve emulation of algorithm steps with neural networks by supervised learning of computational trajectories. Compared to 'one-shot' prediction of algorithm outputs, its sequential nature enables better reasoning and generalisation (Veličković & Blundell, 2021). It has been used primarily for problems of polynomial complexity, with benchmarks such as CLRS-30 (Veličković et al., 2022) driving progress in recent years.

However, several critical limitations of the standard NAR blueprint exist. Firstly, NAR pipelines struggle with ensuring globally valid solutions without significant post-processing and cannot reason about multiple equivalent solutions (Kujawa et al., 2025). Secondly, attempts to apply NAR to NP-hard problems rely on highly specialised approaches demanding significant engineering work (Georgiev et al., 2024a; He & Vitercik, 2025), sacrificing generality in the process. Lastly, NAR is inherently limited to problems where an expert algorithm is available, and thus cannot be applied to discover algorithms for new problems.

The key insight underlying our work is that the NAR blueprint, which breaks down an algorithmic trajectory into a series of steps with a defined feature space, can be viewed as a Markov Decision Process (MDP), the mathematical formalism underpinning reinforcement learning (RL). This correspondence, summarised in Figure 1A, unlocks the use of powerful RL tools for addressing key limitations of NAR: (1) MDP formulations provide a skeleton for ensuring valid solutions by construction and acceptance of several equivalent solutions; (2) we unify polynomial and NP-hard graph problems under a framework that, unlike existing approaches, retains good performance on the latter class while being general; (3) we enable NAR to go beyond known algorithms by implicit learning of new ones using only a reward signal.

Our contributions are as follows: (1) we propose the Graph Neural Algorithmic Reasoning with Reinforcement Learning (**GNARL**) framework that relies on the insights listed above to reframe NAR as an RL problem; (2) we build a general learning architecture that is broadly applicable for graph problems in both P and NP; (3) we carry out an extensive evaluation demonstrating that GNARL can construct valid solutions without pre-processing, achieves comparative or better performance

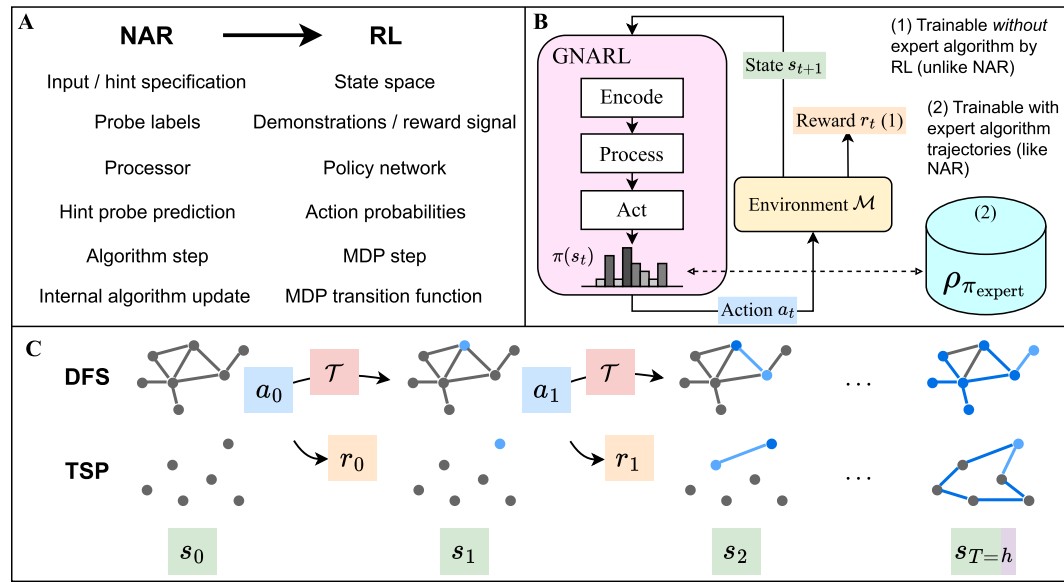

Figure 1: **A.** Key correspondences leveraged by GNARL to cast NAR as an RL problem. **B.** Unlike standard NAR, GNARL is trainable without an expert algorithm by using a reward signal. **C.** Examples of the MDP $\mathcal{M} = \langle\, S\,,\, \mathcal{A}\,,\, \mathcal{T}\,,\, R\,,\, h\, \rangle$ for a polytime solvable and NP-hard problem. At each step, a node is selected, the transition function yields the next state, and a reward is obtained.

on NP-hard problems relative to existing problem-specific NAR methods, and is applicable to new problems even in the absence of an expert algorithm.

## 2  RELATED WORK

### 2.1  NEURAL ALGORITHMIC REASONING

Neural Algorithmic Reasoning (NAR) is a field introduced by Veličković & Blundell (2021) that targets learning to execute algorithms using neural networks. Unlike approaches that learn direct input-output mappings, NAR models are trained on intermediate steps (*trajectories*) of algorithms, ultimately achieving stronger reasoning and out-of-distribution (OOD) generalisation (Veličković et al., 2020; Mahdavi et al., 2023). NAR has advantages over traditional algorithms when handling real-world data, as it can integrate priors to process high-dimensional, noisy, and unstructured data, generalising beyond low-dimensional inputs (Deac et al., 2021; Panagiotaki et al., 2025).

To date, most applications of NAR have been on algorithms of polynomial complexity (P), predominantly those in the CLRS-30 Benchmark (Veličković et al., 2022; Ibarz et al., 2022). The supervised approach in early NAR works (Veličković et al., 2022; 2020) is limited by the requirement for ground-truth labels, inherently restricting training for NP-hard problems to small examples or algorithmic approximation. Recent methods using NAR with self-supervised learning (Bevilacqua et al., 2023; Rodionov & Prokhorenkova, 2023), transfer learning (Xhonneux et al., 2021), fixed point methods (Xhonneux et al., 2024; Georgiev et al., 2024b), and reinforcement learning (Deac et al., 2021) focus on problems in P. Attempts to apply NAR to NP-hard tasks have been restricted to specific problems (He & Vitercik, 2025), or are outperformed by simple heuristics (Georgiev et al., 2024a). Furthermore, these approaches do not guarantee a valid solution, requiring augmentations such as extensive beam search at runtime to ensure valid outputs.

For many problems, there are multiple correct solutions, e.g., as in depth-first search (DFS). However, the CLRS-30 Benchmark relies on a single solution induced by a pre-defined node order. The typical metric reported for NAR models is the *node accuracy*, or micro-$F_1$ score, which corresponds to the mean accuracy of predicting the label of each node (Veličković et al., 2022). Even though recent NAR models achieve high node accuracy (Veličković et al., 2022; Bevilacqua et al., 2023;

Bohde et al., 2024), this metric ignores the fact that a single incorrect prediction can completely invalidate a solution; for a shortest path problem, one incorrect predecessor label could render all paths infinite. Graph accuracy instead measures the percentage of *graphs* for which *all* labels are correctly predicted (Minder et al., 2023). Notably, NAR models score extremely poorly in this more realistic metric, with Kujawa et al. (2025) struggling to achieve beyond 50% for OOD graph sizes.

## 2.2 REINFORCEMENT LEARNING FOR GRAPH COMBINATORIAL OPTIMISATION

Machine learning approaches to combinatorial optimisation have gained popularity in recent years. The goal is to replace heuristic hand-designed components with learned knowledge in the hope of obtaining more principled and optimal algorithms (Bengio et al., 2021; Cappart et al., 2023). This area is also referred to as Neural Combinatorial Optimisation (NCO). Particularly relevant to this paper are works that use reinforcement learning to automatically discover, by trial-and-error, heuristic solvers that generate approximate solutions (Darvariu et al., 2024).

The problems treated in NCO are NP-hard or, even when lacking a formal complexity characterisation, clearly computationally intractable. This approach has mainly been applied to canonical problems such as the Travelling Salesperson Problem (Kwon et al., 2020), Maximum Cut (Khalil et al., 2017), and Maximum Independent Set (Ahn et al., 2020). With few exceptions, the performance of RL-discovered solvers still lags behind traditional ones. The case for RL becomes stronger for problems that currently lack powerful solvers such as Robust Graph Construction (RGC) (Darvariu et al., 2021a) or Molecular Discovery (You et al., 2018), for which an RL-discovered solver is able to achieve high-quality results compared to simple heuristics. Furthermore, Yehuda et al. (2020) highlight that polytime samplers cannot solve NP-hard problems optimally when trained using supervised learning, allowing only approximate solutions. This analysis does not apply to RL, further motivating its use for NP-hard problems.

We highlight that the goal of our work is not to compete directly with NCO methods, which are typically built and optimised for specific problems. Instead, GNARL is designed, in the spirit of NAR, to be a general-purpose framework with straightforward application to new problems. We aim to address the key limitations of NAR and render it applicable for combinatorial optimisation problems, with which the framework has historically struggled. GNARL also makes significant progress towards the challenge indicated by Cappart et al. (2023), who argue that integrating graph neural networks for combinatorial optimisation requires a framework that abstracts technical details.

## 3 BACKGROUND

### 3.1 NEURAL ALGORITHMIC REASONING

In NAR, a ground-truth algorithm A (e.g. DFS, Bellman-Ford) generates a sequence of intermediate steps $\{\mathbf{y}^{(t)} = \mathbb{A}(G_t)\}_{t=0}^{T}$, with final output $\mathbf{y}^{(T)} = \mathbb{A}(G_T)$. Here $G_t = (V_t, E_t)$ corresponds to the input graph at each step $t$, generated by the algorithmic execution at the previous step, with $V_t$ and $E_t$ denoting the sets of nodes and edges, and $\mathbf{x}_v^{(t)}$, $v \in V_t$ and $\mathbf{e}_{uv}^{(t)}$, $(u, v) \in E_t$, their associated features. Instead of training on the final output to learn a direct input-output mapping $f : G_0 \to \mathbb{A}(G_T)$, NAR also uses the intermediate steps (i.e., *hints*) as supervision signals to learn the entire algorithmic *trajectory*. An important and non-trivial aspect of this framework is finding the right hints, which should contain enough information to guide the model towards correctly approximating the algorithmic trajectory, while avoiding unnecessary complexity.

The standard model architecture in NAR approaches relies on the iterative application of Graph Neural Networks (GNNs) with intermediate supervision signals typically following the 'encode-process-decode' paradigm (Hamrick et al., 2018). In this paradigm, input features are first encoded by the network, $\mathbf{z}_v^{(t)} = \text{enc}_V(\mathbf{x}_v^{(t)})$ and $\mathbf{z}_{uv}^{(t)} = \text{enc}_E(\mathbf{e}_{uv}^{(t)})$, then passed through a message-passing neural network (MPNN) (Gilmer et al., 2017) that iteratively updates latent features, and finally decoded into outputs. At each iteration the latent features $\mathbf{h}^{(t)}$ are updated as follows:

$$\mathbf{m}_v^{(t)} = \sum_{u \in \mathcal{N}(v)} M_t[\mathbf{h}_v^{(t-1)} \parallel \mathbf{z}_v^{(t)}, \ \mathbf{h}_u^{(t-1)} \parallel \mathbf{z}_u^{(t)}, \ \mathbf{z}_{uv}^{(t)}], \quad \mathbf{h}_v^{(t)} = U_t(\mathbf{h}_v^{(t-1)} \parallel \mathbf{x}_v^{(t)}, \ \mathbf{m}_v^{(t)}), \quad (1)$$

where each node $v$ receives messages from its neighbours $\mathcal{N}(v)$, $M_t$ is a learnable message function, $U_t$ is a learnable update function, and $\parallel$ denotes concatenation. Finally, the decoder maps the embeddings $\mathbf{h}_v^{(t)}$ to outputs $\hat{\mathbf{y}}^{(t)}$. At each iteration, the decoded output from the current step $t$ becomes the input for the next one at $t+1$, until the sequence of $T$ iterations has been completed. NAR relies on step-wise supervision by aligning $\hat{\mathbf{y}}^{(t)}$ with the internal states of the algorithm, and final-task supervision by aligning $\hat{\mathbf{y}}^T$ with the final output $\mathbf{y}^T$.

## 3.2 MARKOV DECISION PROCESSES AND SOLUTION METHODS

A Markov Decision Process is a tuple $\mathcal{M} = \langle S, \mathcal{A}, \mathcal{T}, R, h \rangle$, where i) $S$ is a set of states; ii) $\mathcal{A}$ is the set of all actions, and $\mathcal{A}(s) \subseteq \mathcal{A}$ is the set of available actions in state $s$; iii) $\mathcal{T} : S \times \mathcal{A} \times S \to [0, 1]$ is the transition probability function; iv) $R : S \times \mathcal{A} \to \mathbb{R}$ is the reward function; and v) $h$ is the horizon. A solution to an MDP is a policy $\pi$ mapping each state $s \in S$ to a probability distribution over actions, i.e. $\pi : S \to \Delta(\mathcal{A})$ where $\Delta(\mathcal{A})$ denotes the probability simplex over $\mathcal{A}$. The optimal policy $\pi^*$ maximises the expected cumulative reward, i.e. $\pi^* = \arg\max_\pi \mathbb{E}_\pi \left[ \sum_{t=1}^{h} R(s_t, a_t) \,\middle|\, a_t \sim \pi(\cdot \mid s_t) \right]$.

Reinforcement learning is an approach for solving MDPs using trial-and-error interactions with an environment to learn an optimal policy. The environment is modelled by the MDP $\mathcal{M}$, and represents the world in which the agent acts, producing successor states and rewards when the agent performs an action. RL algorithms aim to improve a policy $\pi$ by taking actions in the environment and using the reward as a feedback signal to update the policy. Popular reinforcement learning algorithms include Q-learning, policy gradient methods, and actor-critic methods (Sutton & Barto, 2018). Actor-critic methods simultaneously learn an actor policy $\pi : S \to \Delta(\mathcal{A})$, and a critic function $\delta : S \to \mathbb{R}$, which estimates the value of a given state. Proximal Policy Optimisation (PPO) (Schulman et al., 2017) is a popular method, using a clipped surrogate objective which updates the actor policy while ensuring that the updated policy does not deviate too much from the previous one. At each iteration, PPO samples experience tuples $\langle s_t, a_t, r_t, s_{t+1} \rangle$ using the current policy $\pi_i$, and the advantage estimator $\hat{A}(s_t, a_t)$ uses $\delta_i$ to estimate how much better the action $a_t$ is compared to the average action in state $s_t$. The critic is updated by minimising the temporal difference error

$$\mathcal{L}_\delta = \mathbb{E}_{(s_t, a_t, r_t, s_{t+1}) \sim \rho_{\pi_i}} \left[ (r_t + \gamma \delta(s_{t+1}) - \delta(s_t))^2 \right], \tag{2}$$

where $\rho_{\pi_i}$ is the distribution induced by $\pi_i$.

Imitation learning (IL) is another solution method for MDPs. Instead of updating the policy using a loss derived from the rewards, the actor network is trained to imitate an expert policy. Behavioural Cloning (BC) is the simplest method of IL, which learns a policy directly from state-action pairs or action distributions gathered by executing the expert policy. If the expert policy's full action distribution $\pi_{\text{expert}}(\cdot \mid s)$ is available, the actor network is trained by minimising the Kullback-Leibler (KL) divergence between the predicted and expert distributions, given by $\mathcal{L}_{\pi KL}$. If only state-action pairs are available, the actor is trained by minimising cross-entropy loss $\mathcal{L}_{\pi CE}$:

$$\mathcal{L}_{\pi KL} = \mathbb{E}_{s \sim \rho_{\pi_{\text{expert}}}} \left[ D_{KL} \left( \pi(\cdot \mid s) \parallel \pi_{\text{expert}}(\cdot \mid s) \right) \right], \quad \mathcal{L}_{\pi CE} = -\mathbb{E}_{(s,a) \sim \rho_{\pi_{\text{expert}}}} \left[ \log \pi(a \mid s) \right]. \tag{3}$$

BC can also be used to pre-train an RL policy (Nair et al., 2018). In this case, the critic can be trained alongside the actor using the rewards from the environment.

## 4 NEURAL ALGORITHMIC REASONING AS REINFORCEMENT LEARNING

Our method relies on modelling algorithms as MDPs, transforming the algorithmic reasoning problem from predicting output features to predicting a series of actions. To do so, the *Markov property* must hold: the future state of the problem depends only on the *current* state and action. Many classic polynomial algorithms can be straightforwardly modelled to have the Markov property, as was also recognised by Bohde et al. (2024). Similarly, the execution of combinatorial optimisation algorithms on graphs can typically be framed as sequences of choices of nodes or edges, as performed by the works reviewed in Section 2.2. Using an MDP framing, we unify the previously distinct paradigms of learning trajectories of algorithms for polynomial time solvable and NP-hard combinatorial optimisation problems into a single task of learning a policy over graph element selections. This permits the use of the same learning architecture, and can be trained with or without an expert algorithm.

## 4.1 MDP FORMULATION

We define a graph algorithm MDP $\mathcal{M}_\mathtt{A}$ that models the execution of an algorithm $\mathtt{A}$ on a graph $G = (V, E)$. A graph algorithm $\mathtt{A}$ operates over *input features* from an input space $\mathcal{I}$, which describe the problem instance, and *state features* from a feature space $\mathcal{F}$, which represent the algorithm's internal state and are modified during execution. Additionally, each feature can be assigned to a *location*: node, edge, or graph. In NAR, $\mathcal{I}$ and $\mathcal{F}$ correspond to the input probes and hint probes respectively. We assume that the output of the algorithm corresponds to a subset of the state features.

In general, a graph algorithm is naturally framed as a sequential selection of nodes or edges, upon which an operation is applied. Aligning this concept with the MDP action, we consider the core action of $\mathcal{M}_\mathtt{A}$ to be the selection of a node $v \in V$. If a graph algorithm operates over edges, the edge can be constructed by selecting nodes in two consecutive *phases*. Accordingly, we define a graph feature $p$ in each state of $\mathcal{M}_\mathtt{A}$ which reflects the current phase of the action selection. The maximum number of phases $\mathcal{P}$ is given by the problem, with values of 1, 2, and 3 corresponding to algorithms operating on nodes, edges, and triangles respectively. To satisfy the Markov property, we additionally define a set of node state features $\psi_p$, representing the node selected in phase $p$.

Formally, we define the $\mathcal{M}_\mathtt{A}$ as follows. **States.** The states of $\mathcal{M}_\mathtt{A}$ correspond to the input features and state features: $S = \mathcal{I} \times \mathcal{F}$. This includes the phase feature $p$ and the previous node features $\psi_p$ required by the architecture, as well as all of the problem specific features required to represent the algorithm. **Actions.** The action space $\mathcal{A}$ is defined as the set of nodes $V$, and the actions available in each state $\mathcal{A}(s)$ are specified for each problem according to the problem structure. **Transitions.** The transition function $\mathcal{T}$ is defined by the algorithm being executed, and generally corresponds to the fundamental internal update of the algorithm. Appendix B provides further discussion on defining the transition function. **Rewards.** Suppose we wish to maximise an objective function $J : S \to \mathbb{R}$ in the terminal state of the MDP. We set the reward function $R = J(s') - J(s)$ for $s \in S \setminus s_0$, $R(s_0) = 0$. By the reward shaping theorem (Ng et al., 1999), the policy that maximises this reward function also maximises the MDP with the reward only in the terminal state. **Horizon.** The horizon $h$ is defined by the problem, according to the worst-case number of steps. Unlike NAR where the number of processor steps depends on the pre-determined quantity of hints or a separate termination network (Veličković et al., 2022), the horizon is independent of the trajectory.

As an example, we provide the MDP formulation for the DFS algorithm shown in Figure 1C, with others described in Appendix C. The input and state features can be found in Table 1, with types defined as per the CLRS-30 Benchmark. The features used are a simplification of the hints used in the Benchmark, with the addition of the phase and last selected features. The horizon $h = (|V| - 1)\mathcal{P}$. The transition function $\mathcal{T}$ is described in Algorithm 1, where $v$ is the selected action. The available actions are $\mathcal{A}(s) = V$ when $p = 1$ (all nodes), and $\mathcal{A}(s) = \{v | (\psi_1, v) \in E\}$ when $p = 2$ (outgoing edges of the previously selected node). For DFS, we train GNARL using IL only, and therefore a reward function is not required.

---

**Algorithm 1:** DFS Transitions $\mathcal{T}$

$p \leftarrow 1$
**Function** STEPSTATE$(v)$
    **if** $p = 2$ **then**
        $\texttt{reach}_{\psi_1} \leftarrow 1$
        $\texttt{reach}_v \leftarrow 1$
        $\texttt{pred}_v \leftarrow \psi_1$
    $\psi_p \leftarrow v$
    $p \leftarrow p \bmod \mathcal{P} + 1$

---

Table 1: Features for the DFS algorithm.

| Feature | Description | Stage | Location | Type | Initial Value |
|---|---|---|---|---|---|
| adj | Adjacency matrix | Input | Edge | Scalar | - |
| $p$ | Phase ($\mathcal{P} = 2$) | State | Graph | Categorical | 1 |
| $\psi_m$ for $m = 1, \ldots, \mathcal{P}$ | Node last selected in phase $m$ | State | Node | Categorical | $\emptyset$ |
| pred | Predecessor in the tree | State | Node | Pointer | $v \forall v \in V$ |
| reach | Node has been searched | State | Node | Mask | $0 \forall v \in V$ |

## 4.2 ARCHITECTURE

Figure 2 shows the architecture of the GNARL framework, with each stage highlighted. We replace the 'encode-process-decode' paradigm from NAR with *encode-process-act*. The encode and process

stages reflect their NAR counterparts, but in the final stage we transform the processed features into the *action probability* space instead of the input space.

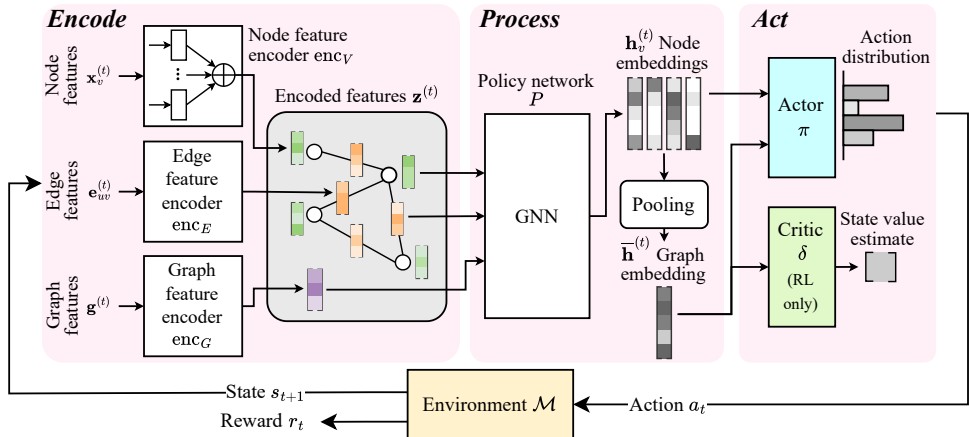

Figure 2: Architecture of the GNARL framework. Each state and input feature is encoded separately, and aggregated by feature location. The processor embeds the state features into a proto-action space. The actor calculates node probabilities using the similarity of the graph embedding to the proto-action vectors. The critic uses the graph embedding to estimate the state value.

**Encoder.** We employ an encoding process similar to Ibarz et al. (2022). At each step, for each distinct input and state feature, a linear transform maps the feature into the embedding space with dimension $f = 64$. The transformed features are then aggregated with other features of the same location (node, edge, or graph). This produces a graph encoding given by $\{\mathbf{z}_v^{(t)}, \mathbf{z}_{uv}^{(t)}, \mathbf{z}_g^{(t)}\}$, with shapes $n \times f$, $n \times n \times f$, and $f$ for node, edge, and graph features respectively. To maintain the Markov property, we encode all input and state features at every step.

**Processor.** The encoded features are fed into a GNN processor $P$, which performs $L$ rounds of message passing to calculate node embeddings $\mathbf{h}_v^{(t)}$. These node embeddings are passed to a pooling layer which calculates an embedding for the graph $\overline{\mathbf{h}}^{(t)} = \text{pool}_{v \in V}(\mathbf{h}_v^{(t)})$. Bohde et al. (2024) find that removing the passing of latent encodings between NAR iterations achieves better algorithmic alignment with the Markov property and produces better performance. Thus, for the processor, we use a modified version of the MPNN (Veličković et al., 2022) and TripletMPNN (Ibarz et al., 2022) in which the latent embeddings are removed, i.e., we omit $\mathbf{h}^{(t-1)}$ in Equation (1).

**Actor.** The actor network takes the node and graph embeddings as input and outputs a probability distribution over the actions. As the policy must be flexible w.r.t. graph size, we adapt the proto-action approach of Darvariu et al. (2021b). The proto-action is computed by applying a learned linear transform $\Theta$ to the graph embedding, further described in Appendix A.1. The similarity sim between each node embedding and the proto-action is calculated using an Euclidean metric. We obtain a probability distribution over the actions using the softmax operator: $\pi(a_v|s) = \frac{\exp(\text{sim}_v/T)}{\sum_{u \in \mathcal{V}} \exp(\text{sim}_u/T)}$, where $T$ is a learned temperature and $\text{sim}_v = -\|\mathbf{h}_v^{(t)} - \Theta(\overline{\mathbf{h}}^{(t)})\|_2$.

### 4.3 TRAINING

GNARL can be trained using both IL and RL. In problems with a clear algorithmic prior, such as those in the CLRS-30 Benchmark, we train using BC which closely parallels supervised learning, given the loss functions in Equation (3). Conversely, for many NP-hard problems there may not be a clear expert to imitate, or the expert may not be able to scale to large enough problems to train the model. In this case, training using RL means there is no reliance on an expert algorithm, overcoming a major limitation of existing NAR works. When training using RL, we use PPO (Schulman et al., 2017), which requires a critic module. For the critic network, we use an MLP which takes as input the graph embedding and outputs a scalar state value prediction. We train the critic as described in Section 3.2. When training using only BC, the critic module and reward func-

tion are not needed. However, if BC is used to pre-train a policy before RL fine-tuning, the critic can be warm-started by training it using Equation (2).

### 4.4 Policy Evaluation, Generating Multiple Solutions, and Action Masking

Recall that GNARL learns a probability distribution $\pi$ over actions. During policy evaluation, actions are chosen greedily (i.e., the highest-probability action is selected). Different solutions can also be reached by *sampling* actions instead. A solution is correct if it could be produced by a deterministic algorithm with some node ordering. GNARL can handle multiple correct solutions better than standard NAR, which requires an arbitrary node ordering (e.g., as encoded in CLRS-30 through the pos feature, which we omit from our setup). Additionally, we use action masking to prevent invalid selections outside of $\mathcal{A}(s)$ such as non-existent edges. This removes the need for problem-specific correction methods, as solutions are valid by construction (though not necessarily optimal).

## 5 Evaluation

In this section, we conduct our evaluation of GNARL over a series of classic graph problems with polynomial algorithms, NP-hard problems, and a problem lacking a strong expert. For brevity, many technical details are deferred to the Appendix.

### 5.1 CLRS Graph Problems

We first evaluate our approach on a selection of graph problems from the CLRS-30 Benchmark (Veličković et al., 2022). The chosen problems present varying levels of difficulty and algorithmic structure. For scalability to large graphs, we use a sparse message passing architecture, as in several works (Minder et al., 2023; Georgiev et al., 2024b). This restricts features to being defined on existing edges only, meaning that not all CLRS-30 graph problems are representable. This is a property of the implementation rather than the GNARL framework itself. For the CLRS-30 problems, we extract distributions over actions from the source algorithm by considering actions under different node orderings as equally valid, and train using BC.

We report the percentage of correctly solved problems in Table 2, which clearly demonstrates the advantage of our approach compared to the vanilla NAR approach (TripletMPNN). Hint-ReLIC (Bevilacqua et al., 2023) and (G-)ForgetNet (Bohde et al., 2024) only report node accuracy, so we estimate the graph accuracy as micro-$F_1^{|V|}$. For algorithms with multiple valid solutions (DFS and BFS), graph accuracy is a strict lower bound for solution correctness, and we provide further analysis in Appendix G.1. Hint-ReLIC makes use of additional hints with reversed pointers which enables 100% OOD node accuracy on DFS, while we use only the original pointers from the CLRS-30 Benchmark. Rodionov & Prokhorenkova (2025) report 100% graph accuracy on the BFS, DFS, and MST-Prim problems. Their unique approach using discrete operators and hard attention facilitates impressive generalisation on the CLRS-30 Benchmark, though it is unclear how well it would perform on combinatorial optimisation problems. Of the baselines, only Kujawa et al. (2025) can produce multiple valid solutions, making it the closest comparison to our framework.

Table 2: Solution correctness % on CLRS-30 Benchmark problems (↑), $|V| = 64$, from 5 different seeds. Italicised results are estimates of graph accuracy (lower bound for BFS and DFS).

|  | TripletMPNN | Hint-ReLIC | G-ForgetNet | Kujawa et al. | GNARL$_{BC}$ |
|---|---|---|---|---|---|
| BFS | **100.0$_{\pm 0.0}$%** | *52.6%* | *97.5%* | - | **100.0$_{\pm 0.0}$%** |
| DFS | 24.2$_{\pm 28.6}$% | *100.0%** | *15.4%* | 3$_{\pm 0}$% | 99.8$_{\pm 0.4}$% |
| Bellman-Ford | 12.4$_{\pm 6.5}$% | *5.4%* | *59.0%* | 54$_{\pm 20}$% | **87.2$_{\pm 8.6}$%** |
| MST-Prim | 2.6$_{\pm 4.2}$% | *0.0%* | *4.3%* | - | **39.4$_{\pm 14.7}$%** |

#### 5.1.1 Finding Multiple Solutions

Using the action probabilities output by GNARL, we can explore the ability of the model to find multiple valid solutions for problems with non-unique solutions, such as BFS and DFS.

Given the distribution, we sample an action at each step with a temperature parameter $\lambda$: $\pi_\lambda(a|s) \propto \pi(a|s)^{1/\lambda}$. When $\lambda \to 0$, the action with the highest probability is always chosen, as in Table 2, while when $\lambda \to \infty$ actions are chosen uniformly at random from the available actions.

Figure 3 demonstrates the number of unique solutions found for BFS and DFS on a single graph by sampling actions from the trained models for 100 episodes. Even with temperatures very close to zero, all of the solutions found are unique. For $\lambda = 0.05$, BFS finds 100 unique solutions in 100 runs, while DFS has an 80% success rate with each solution being unique. With higher temperatures, the success rate decreases, as more incorrect actions are sampled.

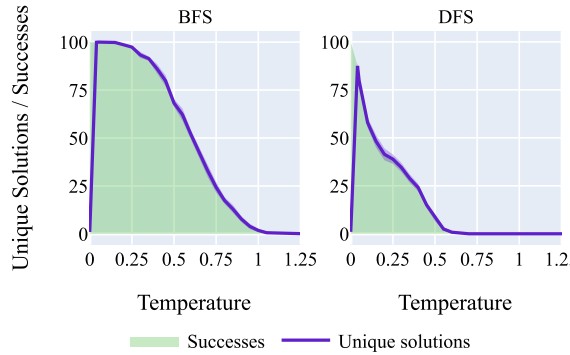

Figure 3: Unique solutions in 100 runs found on a single graph, $|V| = 64$, using sampled actions (10 seeds).

## 5.2 TRAVELLING SALESPERSON PROBLEM

The Travelling Salesperson Problem (TSP) is a thoroughly studied combinatorial optimisation problem, allowing us to evaluate the performance of GNARL with an optimal baseline. Previously, the problem has been approached by using NAR to predict the probability of each node being a node's predecessor in a tour, then extracting valid tours using beam search (Georgiev et al., 2024a). We demonstrate that we can produce valid-by-construction solutions, alleviating the need for beam search, and achieve superior performance using both IL and RL as training methods.

We model the TSP using a constructive approach in which each action selects the next node in the tour, terminating when all nodes have been selected. Only nodes not already in the tour can be selected, as actions are restricted to $\mathcal{A}(s) = \{v \mid \text{in\_tour}_v = 0\}$, thus all solutions are valid by construction. We use the same input features as Georgiev et al. (2024a), and train using 10% of their training data. We first train a policy using BC, with demonstrations generated from the optimal solutions provided by the Concorde solver (Applegate et al., 1998). We also train a policy *without* expert trajectories, relying only on the optimisation of the reward function via PPO.

Results in Table 3 show the performance of GNARL trained with BC and GNARL trained with PPO as compared to the best result of all methods presented by Georgiev et al. (2024a), and against the handcrafted Christofides heuristic (Christofides, 1976). These results clearly demonstrate that GNARL scales effectively using both methods, performing similarly to the heuristic for graphs up to 5 times the training size. We outperform Georgiev et al. (2024a) in OOD generalisation, despite using a fraction of the training data and not using beam search. Further investigations on pre-training using a weak expert can be found in Appendix G.2.

Table 3: TSP percentage worse than optimal objective for OOD graph sizes ($\downarrow$). Non-NAR baselines are italicised. Additional baselines can be found in Georgiev et al. (2024a).

| | **Test size** | | | | | |
| Model | 40 | 60 | 80 | 100 | 200 | 1000 |
| --- | --- | --- | --- | --- | --- | --- |
| *Christofides* | $10.1_{\pm 3}$ | $11.0_{\pm 2}$ | $11.3_{\pm 2}$ | $12.1_{\pm 2}$ | $12.2_{\pm 1}$ | $12.2_{\pm 0.1}$ |
| Georgiev et al. | $7.5_{\pm 1}$ | $13.3_{\pm 3}$ | $19.0_{\pm 3}$ | $23.8_{\pm 5}$ | $33.6_{\pm 3}$ | $38.5_{\pm 3}$ |
| GNARL$_{BC}$ | $7.5_{\pm 0.4}$ | $9.9_{\pm 0.3}$ | $10.9_{\pm 0.6}$ | $12.1_{\pm 0.6}$ | $14.5_{\pm 0.4}$ | $18.6_{\pm 1.0}$ |
| GNARL$_{PPO}$ | $8.1_{\pm 0.4}$ | $9.9_{\pm 0.5}$ | $11.7_{\pm 1.1}$ | $13.1_{\pm 1.2}$ | $15.8_{\pm 1.3}$ | $20.0_{\pm 2.2}$ |

We note that our MDP framing aligns with the only other existing NAR work addressing the TSP to enable a fair comparison. However, other modelling choices may lead to better approximation ratios. For example, in the NCO literature, Khalil et al. (2017) use a helper function in $\mathcal{T}$ that determines the best insertion position for a node instead of mandating insertions be made at the head. GNARL is sufficiently flexible to allow different MDP models to be explored in future work.

## 5.3 Minimum Vertex Cover

The Minimum Vertex Cover (MVC) is a classic NP-hard problem with broad applicability, and has previously been approached using NAR in He & Vitercik (2025). Their PDNAR method uses a dual formulation of the problem to create a bipartite graph for training using both an approximation algorithm and the optimal solution, followed by a clean-up stage.

We model the MVC MDP as a sequential selection of nodes comprising the vertex cover. As a baseline, we compute the $2/(1 - \epsilon)$-approximation from Khuller et al. (1994), with $\epsilon = 0.1$. We use the training data from He & Vitercik (2025), consisting of $10^3$ Barabási-Albert (BA) graphs with $|V| = 16$, and we validate and test on their data. For BC we train using expert demonstrations generated by an exact ILP solver, and for PPO we train directly on the reward function.

Results in Table 4 show the performance of different models as a ratio of the performance of the approximation algorithm. We include PDNAR$_{\text{No algo}}$, which is an ablation of PDNAR using only information from the optimal solution, the same data provided to GNARL. We significantly outperform the directly comparable variant of PDNAR and the approximation algorithm at large scales relative to the training size. GNARL achieves similar in-distribution performance to the full PDNAR without relying on approximation algorithm trajectories, but degrades less gracefully as instance size increases. In summary, our general method obtains comparable results to PDNAR without requiring a dual formulation, approximation algorithm, or refinement stage, greatly simplifying both the modelling and solution pipeline.

Table 4: Model-to-algorithm ratio of objective functions ($J/J_{\text{approx}}$) for MVC ($\downarrow$). Models trained directly on the approximation algorithm steps are italicised. Averaged from 5 different seeds.

| $J/J_{\text{approx}}$ | Test size | | | | | | |
|---|---|---|---|---|---|---|---|
| | 16 (1×) | 32 (2×) | 64 (4×) | 128 (8×) | 256 (16×) | 512 (32×) | 1024 (64×) |
| *TripletMPNN* | $0.982_{\pm 0.015}$ | $0.986_{\pm 0.015}$ | $0.991_{\pm 0.013}$ | $0.995_{\pm 0.020}$ | $1.000_{\pm 0.013}$ | $1.001_{\pm 0.019}$ | $1.005_{\pm 0.019}$ |
| *PDNAR* | $0.943_{\pm 0.004}$ | $0.957_{\pm 0.002}$ | $0.966_{\pm 0.002}$ | $0.958_{\pm 0.002}$ | $0.958_{\pm 0.002}$ | $0.958_{\pm 0.002}$ | $0.957_{\pm 0.002}$ |
| PDNAR$_{\text{No algo}}$ | $1.142_{\pm 0.038}$ | $1.115_{\pm 0.027}$ | $1.110_{\pm 0.038}$ | $1.099_{\pm 0.032}$ | $1.091_{\pm 0.034}$ | $1.099_{\pm 0.036}$ | $1.095_{\pm 0.038}$ |
| GNARL$_{\text{BC}}$ | $0.948_{\pm 0.002}$ | $0.959_{\pm 0.007}$ | $0.971_{\pm 0.013}$ | $0.978_{\pm 0.020}$ | $0.991_{\pm 0.031}$ | $0.998_{\pm 0.037}$ | $1.011_{\pm 0.046}$ |
| GNARL$_{\text{PPO}}$ | $0.945_{\pm 0.002}$ | $0.968_{\pm 0.008}$ | $0.985_{\pm 0.014}$ | $0.994_{\pm 0.021}$ | $1.002_{\pm 0.023}$ | $1.024_{\pm 0.035}$ | $1.033_{\pm 0.039}$ |

## 5.4 Robust Graph Construction

We also study the robust graph construction (RGC) domain from Darvariu et al. (2021a), in which edges must be added to a graph to improve the robustness against disconnection under node removal. In this problem there is no clear expert, and the objective function itself is expensive to calculate, meaning that a neural approach is highly appropriate. Problems of this nature are common in many practical contexts, yet have not previously been approached in the NAR literature.

We use data from Darvariu et al. (2021a) to evaluate the performance of GNARL in contrast to their specifically designed graph RL model, RNet-DQN. We design state features and the transition function similarly to RNet-DQN, but in a way which aligns with the GNARL framework. All models are trained on a set of Erdős-Renyi (ER) or BA graphs with $|V| = 20$, with the removal strategy being either random or targeted as per Darvariu et al. (2021a). As there is no expert algorithm for this problem, we train using PPO directly for $10^7$ steps. We also investigate the effect of warm-starting training using BC. We consider a weak expert (WE) policy which greedily selects the next edge which maximises the reward function, and fine-tune using PPO (WE+PPO).

Results in Figure 4 demonstrate the performance of each method on different OOD graph sizes. The GNARL approaches are competitive with RNet-DQN, which is to be expected due to the similarities in architecture. Interestingly, for BA graphs under the targeted removal strategy, GNARL trained with PPO improves on the generalisation ability of RNet-DQN. The policy trained only on the weak expert performed very poorly on this class of graphs, though training curves suggest that more training epochs may have mitigated this phenomenon. While a traditional NAR approach is not applicable in this scenario due to the absence of an expert algorithm, GNARL allows us to solve this problem using a specification very similar to those used in NAR.

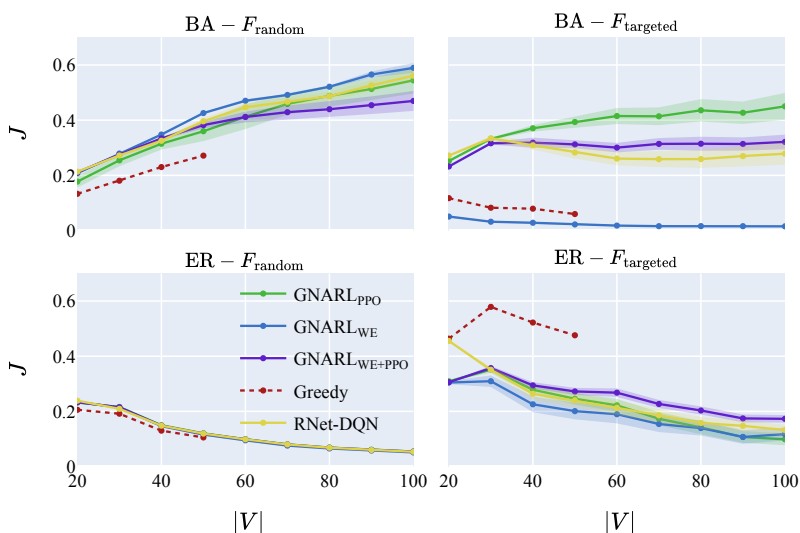

Figure 4: Performance on RGC for different graph sizes ($\uparrow$).

## 6 LIMITATIONS AND FUTURE WORK

There are several important directions for future work. First, defining the MDP presently requires knowledge of the basic steps for constructing a valid solution. In contrast with standard NAR, which can be executed without hints at runtime, it is not possible to execute GNARL without an MDP definition. Nevertheless, the effort required to define the MDP does not appear any more involved than the design of hints for standard NAR. Furthermore, GNARL relies on the environment during execution, creating a performance bottleneck. Both of these limitations might be addressed by using learnable world models (Schrittwieser et al., 2020; Chung et al., 2023) instead of explicitly defined transition functions. Second, when training a model using BC, states outside of $\rho_{\pi_{\text{expert}}}$ are not encountered. During execution, if an action is sampled that transitions to a state outside of $\rho_{\pi_{\text{expert}}}$, the model is not likely to correctly predict the best actions, and the episode often ends in failure. For longer action sequences, the chance of encountering such an action is higher, and as a result the model does not perform as well in problems with longer trajectories. In future work, this could be mitigated using a more advanced imitation learning technique such as DITTO (DeMoss et al., 2025), which brings expert trajectories and policy rollouts closer together in the latent space of a learned world model. This offers a principled and promising way of improving the robustness of GNARL.

## 7 CONCLUSION

In this work, we have presented GNARL, a framework that reimagines the learning of algorithm trajectories as Markov Decision Processes. Doing so unlocks the powerful sequential decision-making tools of reinforcement learning for NAR and serves as a critical bridge between the two largely distinct fields. GNARL addresses the key limitations of classic NAR and is broadly applicable to problems spanning those solvable by polynomial algorithms, as well as challenging combinatorial optimisation problems. Our approach can natively represent multiple correct solutions and removes the need for inference-time repair procedures, performs the same or better than narrow NAR methods on NP-hard problems, and can be applied even when an expert algorithm is missing entirely. In our view, this work is an important step towards achieving a combinatorial optimisation framework that abstracts from technical details, as envisaged by Cappart et al. (2023).

### REPRODUCIBILITY STATEMENT

All code required to reproduce the results of the paper will be made available in the supplementary materials. Code will be released publicly upon acceptance of the paper. Experiments were

implemented using PyTorch Geometric (Fey & Lenssen, 2019), Gymnasium (Towers et al., 2024), Stable Baselines 3 (Raffin et al., 2021), and CLRS-30 (Veličković et al., 2022) libraries. MPNN and TripletMPNN implementations were taken from Georgiev et al. (2024b). Each training run was executed on a single core of an Intel Platinum 8628 CPU with 4GB of memory using CentOS Linux 8.1. Reproducing the results of the paper for every domain requires approximately 2,000 single-core CPU hours per seed.

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

# A    ARCHITECTURE DETAILS

## A.1    ACTOR NETWORK

Further detail of the actor network including the proto-action mechanism is provided in Figure 5. The action distribution is computed by comparing each node embedding to a proto-action vector, which represents the desired characteristics of the next node to be selected. The graph embedding is passed to a learned linear transformation to produce the proto-action. This proto-action is then compared to the embedding of each node using a similarity function, in this case negative Euclidean distance. The action distribution is produced by passing the similarity scores through a softmax function with a learned temperature.

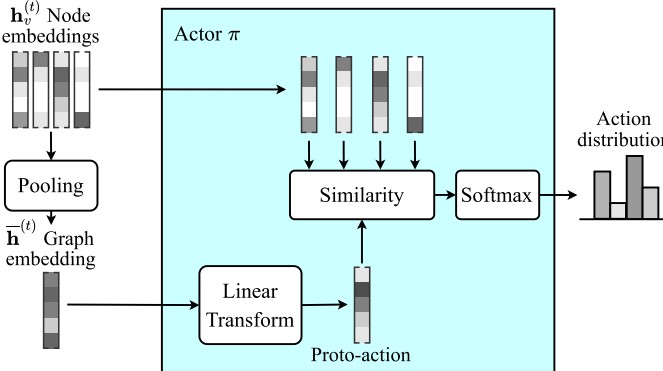

Figure 5: The proto-action is generated from a linear transformation of the graph embedding. This proto-action is then compared to each node embedding using Euclidean distance, which is negated and passed through a softmax function to produce the action distribution.

# B    OBTAINING AN MDP FROM AN ALGORITHM

## B.1    WORKED EXAMPLE

The goal of the GNARL framework is to learn to execute classical algorithms by modelling them as MDPs. In order to model a classical algorithm as an MDP, we must define a transition function, which represents the way in which the algorithm's state changes when an action is taken. Designing the MDP based on the algorithm is a matter of identifying the decision variables, which become the actions, and the state updates, which become the transition function. We illustrate this process using the Bellman-Ford algorithm, given in Algorithm 2, as defined by the CLRS-30 Benchmark (Veličković et al., 2022). The input, output, and hint features used in the CLRS-30 Benchmark for Bellman-Ford are given in Table 5.

**Algorithm 2:** Bellman-Ford Reference Algorithm

**Input:** Adjacency matrix `adj`, weight matrix $A \in \mathbb{R}^{n \times n}$, start node $v_s$
**Output:** Predecessor array `pred` $\in \{1, \ldots, n\}^n$

1   Obtain $G = (V, E)$ from `adj`
2   $d_v \leftarrow 0 \; \forall v \in V$
3   $\text{pred}_v \leftarrow v \; \forall v \in V$
4   $\text{mask}_v \leftarrow 0 \; \forall v \in V$
5   $\text{mask}_{v_s} \leftarrow 1$

6   **while** True **do**
7     $d_{\text{prev}} \leftarrow d$
8     $\text{mask}_{\text{prev}} \leftarrow \text{mask}$
9     **foreach** $u \in V$ **do**
10       **foreach** $v \in V$ **do**
11         **if** $\text{mask}_{\text{prev}_u} = 1$ **and** $(u, v) \in E$ **then**
12           **if** $\text{mask}_v = 0$ **or** $d_{\text{prev}_u} + A_{u,v} < d_v$ **then**
13             $d_v \leftarrow d_{\text{prev}_u} + A_{u,v}$
14             $\text{pred}_v \leftarrow u$
15           $\text{mask}_v \leftarrow 1$

16     **if** $d = d_{\text{prev}}$ **then**
17       **break**

18   **return** `pred`

Table 5: Features in the CLRS-30 Benchmark for the Bellman-Ford algorithm

| Feature | Description | Stage | Location | Type | Initial Value |
|---|---|---|---|---|---|
| `adj` | Adjacency matrix | Input | Edge | Scalar | Given |
| $A$ | Weight matrix | Input | Edge | Scalar | Given |
| $v_s$ | Start node | Input | Node | Mask One | Given |
| `pred` | Predecessor in the shortest path | Hint/Output | Node | Pointer | $v \; \forall v \in V$ |
| `mask` | Node has been visited | Hint | Node | Mask | $0 \; \forall v \in V$ |
| $d$ | Current best distance | Hint | Node | Float | $0 \; \forall v \in V$ |

The Bellman-Ford algorithm consists of a main loop (Lines 6–17) which continues until no distance updates occur (Line 16). Within this loop, each edge outgoing from visited nodes is relaxed (Lines 11–15). Thus, the fundamental unit of the algorithm is to relax a given edge. We can therefore represent the MDP such that the transition function pertains to the relaxation of a single edge. As the GNARL framework operates over nodes rather than edges, we split the selection of an edge $(u, v)$ into two steps: first selecting the source node $u$, then selecting the target node $v$. Thus, we choose $\mathcal{P} = 2$, and using the features from the CLRS-30 Benchmark, arrive at the state features described in Table 6. Note that the adjacency matrix is not listed as an explicit feature as it can be inferred from the non-zero entries in the weight matrix, $A$.

Table 6: Features for the Bellman-Ford algorithm

| Feature | Description | Stage | Location | Type | Initial Value |
|---|---|---|---|---|---|
| $A$ | Weight matrix | Input | Edge | Scalar | - |
| $v_s$ | Start node | Input | Node | Mask One | - |
| $p$ | Phase ($\mathcal{P} = 2$) | State | Graph | Categorical | 1 |
| $\psi_m$ for $m = 1, \ldots, \mathcal{P}$ | Node selected in phase $m$ | State | Node | Categorical | $\emptyset$ |
| `pred` | Predecessor in the shortest path | State | Node | Pointer | $v \; \forall v \in V$ |
| `mask` | Node has been visited | State | Node | Mask | $0 \; \forall v \in V$ |
| $d$ | Current best distance | State | Node | Float | $0 \; \forall v \in V$ |

All that remains is to define the transition function which operates on the defined state features. This is given by the edge relaxation update in Lines 13–15. Notably, we do not include the conditional checks from Lines 11 and 16 in the transition function, as these must be learned by the model. The resultant MDP transition function is given in Algorithm 3. In this transition function, the edge relaxation (Lines 4–6) is only performed in the second phase (Line 2), after both the source node and target node have been selected. To guard against invalid actions, we require that the edge being relaxed exists in the graph (Line 3). Lines 7 and 8 update the feature for the previously selected node and the current phase respectively, as required to keep the Markov property. Using this process, we have derived an MDP representation of the Bellman-Ford algorithm from its classical definition.

---

**Algorithm 3:** Bellman-Ford Transition Function $\mathcal{T}$

1 **Function** STEPSTATE($v$)
2    **if** $p = 2$ **then**
3      **if** $(\psi_1, v) \in E$ **then**
4        $d_v \leftarrow d_{\psi_1} + A_{u,v}$
5        $\texttt{pred}_v \leftarrow \psi_1$
6        $\texttt{mask}_v \leftarrow 1$

7    $\psi_p \leftarrow v$
8    $p \leftarrow p \bmod \mathcal{P} + 1$

---

## B.2   MDP DESIGN CONSIDERATIONS

In designing the MDP, some considerations must be made for the representation of the algorithm. The choice of representation is not unique, and different design decisions may lead to different MDPs for the same algorithm. In the case of Bellman-Ford, we made the design decision to have single-edge relaxations as the fundamental unit of the MDP. An alternative design could have used the choice of node $u$ from Algorithm 2 Line 9 as the action, and had the transition function relax all outgoing edges from $u$. This would greatly reduce the number of steps required to execute the algorithm, as the increased parallelism would allow multiple edges to be relaxed in a single step. However, this would also increase the complexity of the transition function, as it would need to perform multiple edge relaxations and conditional checks. As the goal is for the model to learn the algorithm, this choice of MDP was not used, as it would offload much of the algorithm's logic to the transition function rather than the model. In this manner, the design of the MDP can affect the difficulty of the learning task.

Another important design consideration is maintaining the Markov property of the MDP. This requires that the state features contain all information necessary to determine the next state given an action. As algorithms operate on their internal state, it should always be possible to define a Markovian representation of the algorithm, being careful to include all relevant algorithm state information. Generally, care should be taken when defining a transition function for an algorithm which relies on implicit ordering information such as for loops, or derived information like the root node of a tree, and it may be necessary to include additional state features to capture this information.

The requirement of a hand-designed MDP representation is a limitation of the GNARL framework, though we see it as no more onerous than the hand-selection of features used by the NAR architecture (Veličković et al., 2022). In future work we hope to address this limitation by using world modelling techniques to learn the MDP representation directly from the algorithm's execution trace.

## B.3   MDP STEPS VS NAR STEPS

A key point of difference between the NAR and GNARL representations of an algorithm is the level of parallelism occurring in each execution step. In a single NAR step, the labels for every node and edge are predicted, which allows multiple updates to occur simultaneously. Conversely in GNARL, each step corresponds to a single node selection action, which allows for generality in modelling but results in less parallelism. For example, in the CLRS-30 Benchmark, the Bellman-Ford algorithm is highly parallelised so that each step represents a relaxation of every edge in the graph. In contrast,

the GNARL implementation of Bellman-Ford relaxes a single edge for every $\mathcal{P} = 2$ steps, leading to a much longer episode length. As discussed in Appendix B.2, this design decision was made to ensure that the entire algorithm process was learned by the model, rather than relying on a complex transition function to perform the majority of the work.

In the NAR architecture, the number of steps for the algorithm is pre-determined and is a part of the input specification. This means that if the model does not predict the correct output within the allotted number of steps, the solution is incorrect. In GNARL, the number of steps is bounded by the horizon $h$, but the episode may terminate earlier if a valid solution is reached. For BFS and DFS the episode length is fixed at $h$, while for Bellman-Ford and MST-Prim the episode length may be shorter if the solution is reached early. On the MST-Prim test set, the expert achieves an average length of 462.84, while the horizon is 8192. For Bellman-Ford, the worst-case complexity is $\mathcal{O}(|V|^3)$, so we set the horizon of each problem to be twice the trajectory length achieved by the expert policy, to allow faster evaluation. On the Bellman-Ford test set, the expert achieves an average length of 446.6.

Table 7 shows the number of steps taken by the NAR and GNARL models for each algorithm on graphs with $|V| = 64$, averaged over 5 models on the test set of 100 graphs. Clearly, while the design decision to use single-node updates in GNARL allows for a simple and general transition function, it comes at the cost of a much larger number of steps compared to NAR. This is exacerbated by models with lower success rates such as MST-Prim, where the generous horizon means that unsuccessful episodes greatly skew the average step count. Interestingly, the number of steps taken by GNARL in successful episodes of Bellman-Ford is lower than the expert's average step count, indicating that the learned model is able to find solutions more efficiently than the expert algorithm in some cases. The impact on inference time is discussed in Appendix F.2. In future work, we intend to address this limitation by exploring methods of modelling parallel action selection in the MDP.

Table 7: Number of steps taken for NAR and GNARL models on graphs with $|V| = 64$.

| Method | BFS | DFS | Bellman-Ford | MST-Prim |
|---|---|---|---|---|
| NAR | $3.39_{\pm 0}$ | $192_{\pm 0}$ | $6.41_{\pm 0}$ | $65_{\pm 0}$ |
| GNARL | $128_{\pm 0}$ | $128_{\pm 0}$ | $446_{\pm 69}$ | $4400_{\pm 1850}$ |
| GNARL (success only) | $128_{\pm 0}$ | $128_{\pm 0}$ | $340_{\pm 86}$ | $456_{\pm 5}$ |
| GNARL (failure only) | $128_{\pm 0}$ | $128_{\pm 0}$ | $913_{\pm 18}$ | $8192_{\pm 0}$ |

## C  ENVIRONMENT DETAILS

### C.1  FEATURES, TYPES, AND STAGES

The GNARL implementation framework inherits many aspects of its specification from the CLRS-30 Benchmark (Veličković et al., 2022). Features are the variables serving as the working space of an algorithm. In the benchmark, the state of the features at a given step is called a probe. Each probe has a stage, location, and type. The location is one of node, edge, or graph. The type defines how the probe is represented, and the CLRS-30 Benchmark defines various possible types, such as scalar, categorical, mask, mask-one, and pointer. In the benchmark, each type corresponds to a different loss function, but this does not apply for GNARL as training is performed on either the action distribution or the state-action-reward tuple. Each probe is initialised as per the CLRS-30 Benchmark unless otherwise stated.

In the CLRS-30 Benchmark, the stage of a probe can be either input, hint, or output. Inputs are encoded once in the first round of inference, while hints act as auxiliary probes for intermediate predictions, trained against reference hints. The output probes are trained separately and use their own loss function. We note that hint probes often represent intermediate values of the outputs.

The GNARL framework uses stages in a slightly different manner. There are two stage types: input and state. The input features correspond to immutable features of the graph that are given to the algorithm, and are sufficient for the expert algorithm to solve the problem. This corresponds to the CLRS-30 input features. Unlike NAR, we encode the input features at every step rather than just the first, to ensure that the Markov property holds in the absence of recurrent features in the architecture.

In GNARL, we use state features to denote features which may be altered by the transition function during an episode. This is analogous to the hints in NAR. At each step during execution, both the input features and state features are encoded. We do not use a distinct output feature, and instead consider the output of the algorithm to be some subset of the state features. The evaluation of GNARL relies on the terminal state and/or the reward function, so there is no need to use a separate output feature.

## C.2 BFS / DFS

The BFS algorithm uses the state features in Table 8. Notably, these are the same as for DFS, with the addition of a start node $v_s$ for BFS. The state transition function for both algorithms is given by Algorithm 1. Note that DFS supports directed graphs, while BFS is only run on undirected graphs in the CLRS-30 Benchmark (Veličković et al., 2022). Furthermore, despite these tasks being called *search*, there is no explicit target node that is being searched for, upon which the search can terminate. Instead, the task is to *traverse* the graph and visit all nodes.

We use simple state features as compared to the large number of hints used for DFS in the CLRS-30 Benchmark as they complicate the transition function, whereas we aimed to use the simplest transition function for each environment. The horizon $h = \mathcal{P}(|V| - 1)$. The available actions are $\mathcal{A}(s) = V$ when $p = 1$, and $\mathcal{A}(s) = \{v \mid (\psi_1, v) \in E\}$ when $p = 2$. We do not define an objective function for this problem.

For BFS a solution is considered correct if the depths of the BFS tree are equivalent to a ground-truth BFS tree. For DFS there are many possible solutions given the choice of starting node and subsequent node orderings. We use a recursive approach which confirms that each subtree in a spanning forest is valid, described in Algorithm 4 (Nathaniel, 2025).

For BFS, the expert algorithm outputs action distributions in which all edges at the minimum unvisited depth have equal probability of being selected (Algorithm 9). DFS uses a similar approach, but selects the deepest unvisited node instead (Algorithm 10). We collect 1000 episodes of experience from a set of ER graphs with $|V| \in \{4, 7, 11, 13, 16\}$, where $p_{ER}$ is sampled from $[0.1, 0.9]$. We train for 20 epochs, though in practice the BFS model converged after $< 100$ training steps. For validation we use 100 graphs with $|V| = 16$ and $p_{ER} = 0.5$, and we test on graphs with $|V| = 64$ and $p_{ER} = 0.5$.

Table 8: Features for the BFS algorithm

| Feature | Description | Stage | Location | Type | Initial Value |
|---|---|---|---|---|---|
| $v_s$ (BFS only) | Start node | Input | Node | Mask One | - |
| adj | Adjacency matrix | Input | Edge | Scalar | - |
| $p$ | Phase ($\mathcal{P} = 2$) | State | Graph | Categorical | 1 |
| $\psi_m$ for $m = 1, \ldots, \mathcal{P}$ | Node last selected in phase $m$ | State | Node | Categorical | $\emptyset$ |
| pred | Predecessor in the tree | State | Node | Pointer | $v \, \forall v \in V$ |
| reach | Node has been searched | State | Node | Mask | $0 \, \forall v \in V$ |

## C.3 BELLMAN-FORD

The Bellman-Ford algorithm is implemented using the state features in Table 6 and the transition function in Algorithm 3. The non-phase-related state features are equivalent to the hints used in the CLRS-30 Benchmark. The available actions are $\mathcal{A}(s) = \{v \mid \mathtt{mask}_v = 1\}$ when $p = 1$ and $\mathcal{A}(s) = \{v \mid (\psi_1, v) \in E\}$ when $p = 2$. The horizon $h = \mathcal{P}(|V| - 1)|E|$, but a terminal state may be reached when a correct solution pred is found. A solution is considered correct if each shortest path distance matches a reference solution.

If a reward signal is required, a sensible objective function could be $J = \sum_{v \in V} \min(P_{\mathtt{pred}}(v), |V|)$, where $P_{\mathtt{pred}}(v)$ is the length of the path given by following the predecessor of $v$ as per pred until $v_s$ is reached (assuming normalised edge weights).

For the Bellman-Ford expert demonstrations, we use an algorithm that outputs equal probabilities for all edges of a node being expanded (Algorithm 11), reflecting the parallel expansion used in

---

**Algorithm 4:** Check if a Predecessor Forest is a Valid DFS Solution

---

**Input:** Adjacency matrix `adj`, predecessor array `pred`
**Output:** True if `pred` is a valid DFS forest for `adj`, else False
**Function** `CheckValidDFS(adj, pred)`:
  Obtain $G = (V, E)$ from `adj`
  `active_nodes` $\leftarrow V$
  **return** `IsValidForestRecursive(active_nodes, pred)`

**Function** `IsValidForestRecursive(active_nodes, pred)`:
  **if** $|$`active_nodes`$| \leq 1$ **then**
    **return** True
  `subroots` $\leftarrow \{v \in$ `active_nodes` $\mid$ `pred`$_v \notin$ `active_nodes` or `pred`$_v = v\}$
  **if** `subroots` $= \emptyset$ and `active_nodes` $\neq \emptyset$ **then**
    **return** False
  **if** $|$`subroots`$| > 1$ **then**
    **foreach** $v \in$ `active_nodes` **do**
      `subroot`$_v \leftarrow i$, where $i$ is the root of the subtree containing $v$ by following `pred`
    $G_{\text{component}} \leftarrow$ (`subroots`$, \emptyset$)
    **foreach** $(u, v) \in E$ such that $u, v \in$ `active_nodes` **do**
      **if** `subroot`$_u \neq$ `subroot`$_v$ **then**
        $G_{\text{component}}.E \leftarrow G_{\text{component}}.E \cup \{($`subroot`$_u,$ `subroot`$_v)\}$
    **if** $G_{\text{component}}$ has a cycle **then**
      **return** False
  **foreach** `root` in `subroots` **do**
    `descendants` $\leftarrow$ descendants of `root` in `active_nodes`
    **if** `descendants` $\neq \emptyset$ **then**
      **if** not `IsValidForestRecursive(descendants)` **then**
        **return** False
  **return** True

---

the CLRS-30 Benchmark. Expert experience consists of 10,000 graphs with the same sizes and specifications used for BFS/DFS.

## C.4  MST-PRIM

We implement MST-Prim using the features in Table 9 and the transition function in Algorithm 5. We use fewer state features to the hints in the CLRS-30 Benchmark, omitting the u hint. The horizon $h = \mathcal{P}|V|^2$, but a terminal state may be reached when a correct solution `pred` is found. The available actions are $\mathcal{A}(s) = \psi_1 \cup \{v \mid \text{in\_queue}_v = 1\}$ when $p = 1$ and $\mathcal{A}(s) = \{v \mid (\psi_1, v) \in E\}$ when $p = 2$. A solution is considered correct if the output is a spanning tree and the total weight of the tree is equal to that of a reference solution.

Expert demonstrations are generated using a Markovian implementation of the MST-Prim algorithm provided by the CLRS-30 Benchmark (Veličković et al., 2022), where all edges satisfying the DP equation are assigned an equal probability (Algorithm 12). Expert experience consists of 10,000 graphs with the same sizes and specifications used for BFS/DFS.

Table 9: Features for the MST-Prim algorithm

| Feature | Description | Stage | Location | Type | Initial Value |
|---|---|---|---|---|---|
| $A$ | Weight matrix | Input | Edge | Scalar | - |
| $v_s$ | Start node | Input | Node | Mask One | - |
| $p$ | Phase ($\mathcal{P} = 2$) | State | Graph | Categorical | 1 |
| $\psi_m$ for $m = 1, \ldots, \mathcal{P}$ | Node last selected in phase $m$ | State | Node | Categorical | $\emptyset$ |
| pred | Predecessor in the tree | State | Node | Pointer | $v \, \forall v \in V$ |
| key | Node's key | State | Node | Scalar | $0 \, \forall v \in V$ |
| mark | Node is currently being searched | State | Node | Mask | $0 \, \forall v \in V$ |
| in_queue | Node is in queue | State | Node | Mask | $0 \, \forall v \in V$ |

---

**Algorithm 5:** MST-Prim Transition Function $\mathcal{T}$

---

**Function** STEPSTATE($v$)

    **if** $p = 1$ **then**
        mark$_v \leftarrow 1$
        in_queue$_v \leftarrow 0$

    **if** $p = 2$ **then**
        $u \leftarrow \psi_1$
        **if** $(u, v) \in E$ and mark$_v = 0$ **then**
            **if** in_queue$_v = 0$ or $A_{u,v} <$ key$_v$ **then**
                pred$_v \leftarrow u$
                key$_v \leftarrow A_{u,v}$
                in_queue$_v \leftarrow 1$

    $\psi_p \leftarrow v$
    $p \leftarrow p \bmod \mathcal{P} + 1$

---

## C.5   TSP

**Definition 1** (Travelling Salesperson Problem). *Let $K_n = (V, E)$ be a complete graph of $n$ nodes, with edge weights $w_{(u,v)} \in \mathbb{R}_{>0}$ for $u, v \in V$. Let $T$ be a permutation of $V$ called a tour. The optimal tour of $K_n$ maximises the objective $J = -\sum_{k=1}^{n-1} w_{(T_k, T_{k+1})} + w_{(T_n, T_1)}$.*

For the TSP we use the same input features as Georgiev et al. (2024a), with state features listed in Table 10. As there is only one phase, the phase feature is not strictly necessary, but we include it for consistency with the other environments. The horizon $h = |V|$, as each node must be selected once. The available actions are $\mathcal{A}(s) = \{v \mid \text{in\_tour}_v = 0\}$. To maintain consistency with Georgiev et al. (2024a), we include a starting node in the input features and require this to be the first node selected without loss of generality, so $\mathcal{A}(s_0) = v_s$.

Training data is composed of the first 10% of the graphs used in Georgiev et al. (2024a) for each of the sizes $|V| \in \{10, 13, 16, 19, 20\}$. We use their full validation and test datasets.

Expert demonstrations are created using tours from the Concorde solver (Applegate et al., 1998), where the first selected node is $v_s$, the next node is the node following $v_s$ in the tour, and so on. BC training is performed on 20 epochs of the $5 \times 10^4$ episodes, and PPO training is performed using $10^7$ steps from episodes on randomly sampled training graphs.

---

**Algorithm 6:** TSP Transition Function $\mathcal{T}$

---

**Function** STEPSTATE($v$)

    in_tour$_v \leftarrow 1$
    $u \leftarrow \psi_1$
    pred$_v \leftarrow$ pred$_u$
    pred$_u \leftarrow v$
    $\psi_1 \leftarrow v$

---

Table 10: Features for the TSP

| Feature | Description | Stage | Location | Type | Initial Value |
|---|---|---|---|---|---|
| $A$ | Weight matrix | Input | Edge | Scalar | - |
| $v_s$ | Start node | Input | Node | Mask One | - |
| in_tour | Node in existing partial tour | State | Node | Mask | $0 \, \forall v \in V$ |
| $p$ | Phase ($\mathcal{P} = 1$) | State | Graph | Categorical | 1 |
| $\psi_m$ for $m = 1$ | Node last selected in phase $m$ | State | Node | Categorical | $\emptyset$ |
| pred | Next node in the tour | State | Node | Pointer | $v \, \forall v \in V$ |

## C.6 MVC

**Definition 2** (Minimum Vertex Cover). *Given an undirected graph $G = (V, E)$, a* vertex cover *for $G$ is a set $C \subseteq V$ such that $\forall (u, v) \in E$, at least one of $u, v$ is in $C$. Given a node weight $w_v \in \mathbb{R}_{>0}$ for each $v \in V$, a* Minimum Vertex Cover *is a vertex cover that maximises the objective $J = -\sum_{c \in C} w_c$.*

We model the MVC MDP as a sequential selection of nodes comprising the vertex cover. A solution is valid if each edge has at least one endpoint in the cover. The features used for MVC are found in Table 11 and the transition function is found in Algorithm 7. The horizon $h = |V|$ but a terminal state is entered when the current set of selected nodes forms a valid cover, meaning that most episodes have length $< h$. The available actions are $\mathcal{A}(s) = \{v \mid \text{in\_cover}_v = 0\}$, preventing selection of nodes that are already in the cover and avoiding the need for beam search or a clean-up stage.

Training data is taken from He & Vitercik (2025), consisting of $10^3$ BA graphs with $M \in [1, 10]$ and $|V| = 16$. We generate $10^4$ episodes of experience from this training set, and train for 10 epochs. Validation and test data is taken from the same source, with 100 graphs of $|V| = 16$ for validation and 100 graphs for each test size.

The expert policy is generated from solutions found using an optimal ILP solver, formulated per He & Vitercik (2025). Each unselected node in the optimal solution is assigned an equal probability of being selected next. For BC we train using the expert policy distributions. For PPO we train for $10^7$ steps using episodes on randomly sampled training graphs.

Table 11: Features for MVC

| Feature | Description | Stage | Location | Type | Initial Value |
|---|---|---|---|---|---|
| adj | Adjacency matrix | Input | Edge | Mask | - |
| $w$ | Node weights | Input | Node | Scalar | - |
| in_cover | Node is in the cover | State | Node | Mask | $0 \, \forall v \in V$ |
| $p$ | Phase ($\mathcal{P} = 1$) | State | Graph | Categorical | 1 |
| $\psi_m$ for $m = 1$ | Node last selected in phase $m$ | State | Node | Categorical | $\emptyset$ |

---

**Algorithm 7:** MVC Transition Function $\mathcal{T}$

---

**Function** STEPSTATE($v$)

     $\text{in\_cover}_v \leftarrow 1$

     $\psi_1 \leftarrow v$

---

## C.7 RGC

**Definition 3** (Robust Graph Construction). *Let $G_0 = (V, E_0)$ be a graph and let $\ell < |V|^2 - |E|$ be a positive integer. For $i = 1, \ldots, \ell$, choose a new edge $(u, v) \notin E_{i-1}$, $u, v \in V$, and construct $G_i = (V, E_i)$ where $E_i = E_{i-1} \cup (u, v)$. Define the critical fraction $\xi_G \in (0, 1]$ be the fraction of nodes removed from $G$ in some order until $G$ becomes disconnected. Let $F(G)$ be the expectation of $\xi_G$ under a given removal strategy. Then the objective is to maximise $J = F(G_\ell) - F(G_0)$.*

The features for the RGC environment are found in Table 12, with the transition function shown in Algorithm 8. The horizon corresponds to the edge addition budget $\ell = \lceil 2\tau/(|V|^2 - |V|) \rceil$, where $\tau$, here $\tau = 0.05$, is an input parameter determining the proportion of edges to be added. The available actions are $\mathcal{A}(s) = \{v \mid (u, v) \notin E$ for some $u \in V\}$ when $p = 1$ and $\mathcal{A}(s) = \{v \mid (\psi_1, v) \notin E\}$ when $p = 2$.

We show results for both ER graphs with $p = 0.2$ and BA graphs with $M = 2$. All data is taken from Darvariu et al. (2021a), with training data made up of $10^4$ graphs with $|V| = 20$, validation consisting of 100 graphs with $|V| = 20$, and test data comprising 100 graphs of each size $|V| \in \{20, 40, 60, 80, 100\}$. The weak expert policy is trained using one epoch of demonstrations from the greedy policy for each training graph.

Table 12: Features for RGC

| Feature | Description | Stage | Location | Type | Initial Value |
|---------|-------------|-------|----------|------|---------------|
| adj | Adjacency matrix of current graph | State | Edge | Mask | - |
| $p$ | Phase ($\mathcal{P} = 2$) | State | Graph | Categorical | 1 |
| $\psi_1$ | Node selected in phase $m$ | State | Node | Categorical | $\emptyset$ |
| $\tau_\ell$ | Remaining edge addition budget (fraction) | State | Graph | Scalar | 1 |

---

**Algorithm 8:** RGC Transition Function $\mathcal{T}$

---

**Function** STEPSTATE($v$)

    **if** $p = 2$ **then**

        $u \leftarrow \psi_1$

        $\text{added}_{u,v} \leftarrow 1$

        $\tau_\ell \leftarrow \tau_\ell - 1/(|V|^2 - |V|)$

    $\psi_p \leftarrow v$

    $p \leftarrow p \bmod \mathcal{P} + 1$

---

# D  EXPERT POLICIES

In this section we provide pseudocode for the expert policies used to generate action distributions for the CLRS-30 Benchmark algorithms studied. Each algorithm operates on the current state $s$, which includes the graph $G = (V, E)$ and all node and edge features. The BFS, DFS, Bellman-Ford, and MST-Prim expert algorithms are provided in Algorithms 9–12.

While we implement expert policies that provide the action distribution, it is also possible to implement expert policies that provide a single action demonstration for each state. This is simpler to implement, but requires more training episodes to learn the distribution.

**Algorithm 9: Expert Policy for BFS Environment**

**Function** $\pi^*(s)$:
  **if** $p = 1$ **then**
     **return** `PhaseOnePolicy`$(s)$
  **else**
     **return** `PhaseTwoPolicy`$(s, \psi_1)$

**Function** `PhaseOnePolicy`$(s)$:
  `closed` $\leftarrow \{v \in V \mid \text{reach}_v = 1 \wedge \text{reach}_u = 1 \, \forall u \in \mathcal{N}(v)\}$
  `open` $\leftarrow V \setminus \text{closed}$
  `depths` $\leftarrow$ `GetDepthCounter`$(reach, pred)$
  $d_{\min} \leftarrow \min_{v \in \text{open}} \text{depths}_v$
  `eligible` $\leftarrow \{j \in \text{open} \mid \text{depths}_j = d_{\min}\}$
  **return** $\pi(v) = \begin{cases} 1/|\text{eligible}| & \text{if } v \in \text{eligible} \\ 0 & \text{otherwise} \end{cases}$

**Function** `PhaseTwoPolicy`$(s, \psi_1)$:
  $N \leftarrow \{j \in \mathcal{N}(\psi_1) \setminus \{\psi_1\} \mid \text{reach}_j = 1\}$
  **return** $\pi(v) = \begin{cases} 1/|N| & \text{if } v \in N \\ 0 & \text{otherwise} \end{cases}$

---

**Algorithm 10: Expert Policy for DFS Environment**

**Function** $\pi^*(s)$:
  **if** $p = 1$ **then**
     **return** `PhaseOnePolicy`$(s)$
  **else**
     **return** `PhaseTwoPolicy`$(s, \psi_1)$

**Function** `GetNodeColour`$(reach, adj)$:
  `colour` $\leftarrow \begin{cases} 0 & \text{if } \text{reach}_v = 0 \\ 2 & \text{if } \text{reach}_v = 1 \wedge \forall u \in \mathcal{N}(v), \, \text{reach}_u = 1 \\ 1 & \text{otherwise} \end{cases}$
  **return** `colour`

**Function** `PhaseOnePolicy`$(s)$:
  `colour` $\leftarrow$ `GetNodeColour`$(reach, adj)$
  `depths` $\leftarrow$ `GetDepthCounter`$(colour \neq 0, pred)$
  $d_{\max} \leftarrow \max_{v | \text{colour}_v \neq 2} \text{depths}_v$
  `eligible` $\leftarrow \{v \mid \text{colour}_v = 1 \wedge \text{depths}_v = d_{\max}\}$
  **if** `eligible` $= \emptyset$ **then**
     `eligible` $\leftarrow \{v \mid \text{colour}_v = 0 \wedge \text{depths}_v = d_{\max}\}$
  **return** $\pi(v) = \begin{cases} 1/|\text{eligible}| & \text{if } v \in \text{eligible} \\ 0 & \text{otherwise} \end{cases}$

**Function** `PhaseTwoPolicy`$(s, \psi_1)$:
  $N \leftarrow \{j \in \mathcal{N}(\psi_1) \setminus \{\psi_1\} \mid \text{reach}_j = 0\}$
  **if** $N = \emptyset$ **and** $\text{reach}_{\psi_1} = 0$ **then**
     $N \leftarrow \{\psi_1\}$
  **return** $\pi(v) = \begin{cases} 1/|N| & \text{if } v \in N \\ 0 & \text{otherwise} \end{cases}$

**Algorithm 11:** Expert Policy for Bellman-Ford Environment

**Function** $\pi^*(s)$:
  **if** $p = 1$ **then**
    $\lfloor$ **return** PhaseOnePolicy$(s)$
  **else**
    $\lfloor$ **return** PhaseTwoPolicy$(s, \psi_1)$

**Function** PhaseOnePolicy$(s)$:
  $\text{possible} \leftarrow \begin{cases} \emptyset & \text{if } \exists v \in V, \text{mask}_v = 1 \\ \{v_s\} & \text{otherwise} \end{cases}$
  **foreach** $v \in V$ where $\text{mask}_v = 1$ **do**
    **if** PhaseTwoPolicy$(s, v)$ is not $\emptyset$ **then**
      $\lfloor$ $\text{possible} \leftarrow \text{possible} \cup \{v\}$

  **return** $\pi(v) = \begin{cases} 1/|\text{possible}| & \text{if } v \in \text{possible} \\ 0 & \text{otherwise} \end{cases}$

**Function** PhaseTwoPolicy$(s, u)$:
  $N \leftarrow \{v \in \mathcal{N}(u) \setminus \{u\} \mid \text{dist}_u + A_{uv} < \text{dist}_v \vee \text{mask}_v = 0\}$
  **if** $N = \emptyset$ **then**
    $\lfloor$ **return** $\emptyset$
  **return** $\pi(v) = \begin{cases} 1/|N| & \text{if } v \in N \\ 0 & \text{otherwise} \end{cases}$

---

**Algorithm 12:** Expert Policy for MST-Prim Environment

**Function** $\pi^*(s)$:
  **if** $p = 1$ **then**
    $\lfloor$ **return** PhaseOnePolicy$(s)$
  **else**
    $\lfloor$ **return** PhaseTwoPolicy$(s, \psi_1)$

**Function** PhaseOnePolicy$(s)$:
  **if** $\psi_1$ is set and PhaseTwoPolicy$(s, \psi_1) \neq \emptyset$ **then**
    $\lfloor$ $k \leftarrow \psi_1$               // Maintain current node if possible
  **else**
    $C \leftarrow \{v \in V \mid \text{in\_queue}_v = 1\}$, sorted by increasing key
    **foreach** $u \in C$ **do**
      **if** PhaseTwoPolicy$(s, u) \neq \emptyset$ **then**
        $k \leftarrow u$
        **break**

  **return** $\pi(v) = \begin{cases} 1 & v = u \\ 0 & \text{otherwise} \end{cases}$

**Function** PhaseTwoPolicy$(s, \psi_1)$:
  $N \leftarrow \{v \in \mathcal{N}(\psi_1) \setminus \{\psi_1\} \mid \text{mark}_v = 0 \wedge (\text{key}_v > A_{\psi_1 v} \text{ or } \text{in\_queue}_v = 0)\}$
  **if** $N = \emptyset$ **then**
    $\lfloor$ **return** $\emptyset$
  **return** $\pi(v) = \begin{cases} 1/|N| & v \in N \\ 0 & \text{otherwise} \end{cases}$

# E  TRAINING DETAILS

## E.1  COMPARISON OF PPO AND BC TRAINING

Figure 6 shows the validation set performance of GNARL during training as a function of training time for both BC and PPO on the TSP problem. Here, PPO was trained for $10^7$ episode steps, while BC was trained on 50,000 episodes for 20 epochs ($10^6$ episode steps). PPO reaches a higher performance on the validation set more quickly than BC, but we also see from results in Tables 3, 4 and 19 that policies trained using BC appear to generalise better to larger graphs than those trained using PPO. The investigation of this phenomenon is left to future work.

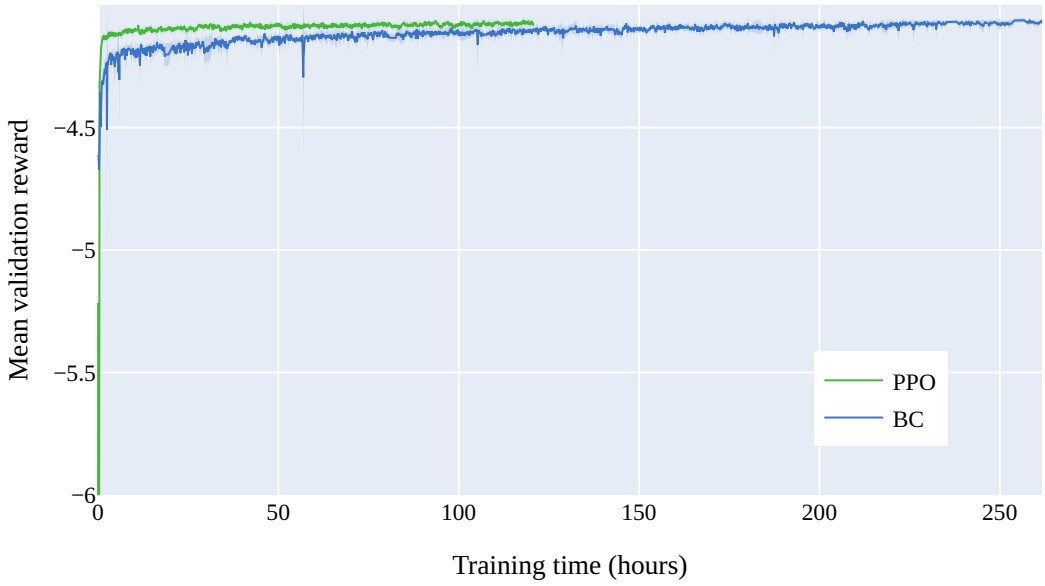

Figure 6: Comparison of BC and PPO training on TSP, averaged over 5 seeds.

## E.2  TRAINING TIME COMPARISON BETWEEN GNARL AND NAR

Table 13 shows the training time for GNARL and the TripletMPNN NAR model (Ibarz et al., 2022) on the CLRS-30 Benchmark algorithms. For the TripletMPNN model, training consists of 10,000 steps of supervised learning using a batch size of 32 graphs. For GNARL, training is performed over a given number of episodes, meaning that the number of steps taken in training depends on the average length of the expert's trajectory (see Appendix B.3). Training was performed on a single core of an Intel Platinum 8628 CPU with 4GB of memory using CentOS Linux 8.1. Due to the difference in implementation frameworks (PyTorch Geometric and Gymnasium for GNARL, and JAX for NAR), a direct comparison is challenging.

Table 13: Training time (hours) for GNARL and NAR models on CLRS-30 Benchmark algorithms.

|              | NAR              | GNARL            |
| ------------ | ---------------- | ---------------- |
| BFS          | $5.25_{\pm0.29}$ | $3.96_{\pm0.19}$ |
| DFS          | $27.3_{\pm2.5}$  | $3.96_{\pm0.06}$ |
| Bellman-Ford | $5.62_{\pm0.11}$ | $107_{\pm1}$     |
| MST-Prim     | $7.57_{\pm0.05}$ | $73.0_{\pm1.7}$  |

## E.3  EFFECT OF CRITIC NETWORK ON TRAINING TIME

When training GNARL using imitation learning, it is not necessary to train the critic network unless the module will later be fine-tuned using an actor-critic method such as PPO. For the TSP prob-

lem, the GNARL model trained using BC for 5000 episodes with the critic enabled took $27.4_{\pm 1.33}$ hours, while training without the critic took $20.1_{\pm 1.06}$ hours (averaged over 5 seeds). The results demonstrate that training the critic network increases training time by approximately 36%.

# F  EVALUATION DETAILS

## F.1  HYPERPARAMETER SEARCH

In order to find the best hyperparameters for the model on different problems, a search of the hyperparameter space was conducted within our computational budget. For each model trained with a single method, a grid search was conducted over the values in Table 14. For the TSP and RGC domains, only MPNN was used. In runs using PPO for fine-tuning, the policy network and PPO parameters were chosen from the best PPO run, and the BC parameters were then selected from the highest scoring BC run with the same policy network parameters.

Table 14: Hyperparameter values used in grid search.

| Property | Values searched |
|---|---|
| *Policy Network* | |
| Processor type | MPNN, TripletMPNN |
| Pooling type | Max, Mean |
| $L$ | 2, 3, 4, 5 |
| *Behavioural Cloning* | |
| Learning rate | 5.0e-2, 1.0e-3, 5.0e-4, 1.0e-4 |
| Batch size | 8, 16, 32, 64, 128 |
| *PPO* | |
| Learning rate | 5.0e-4, 1.0e-5, 5.0e-6 |
| Batch size | 32, 64, 128 |

The aggregation function used in the MPNN was set to Max, given the algorithmic alignment results of Ibarz et al. (2022). However, this causes state aliasing for certain domains where the node and edge features are identical in the initial state, such as RGC and DFS. For these domains, Sum aggregation was used instead.

The final model for evaluation was chosen based on the best achieved validation score, corresponding to mean success for CLRS-30 Benchmark domains and mean reward for NP-hard domains. For runs using PPO this score was taken over the average of 5 different seeds. In case of ties (for example when multiple models achieved mean success of 1), the tie was broken using mean number of steps. For BFS and DFS the number of steps is fixed, so the final hyperparameters were chosen to be the individual parameters which were most common among runs with mean success of 1. The final hyperparameter choices can be found in Table 15.

For all experiments we used $\gamma = 1$. For PPO we used an update interval of 1024 steps, 10 epochs. Other PPO hyperparameters were set according to the defaults in Stable Baselines 3 (Raffin et al., 2021). For BC we ran evaluation every 100 batches. For PPO we ran evaluation at every update. The best model was chosen according to a lexicographical ordering of success rate, mean reward, and mean episode length.

## F.2  CLRS-30 INFERENCE TIME COMPARISON

Table 16 shows the inference time per graph on CLRS-30 Benchmark algorithms using GNARL, and NAR with TripletMPNN architecture (Ibarz et al., 2022). The times were measured on a single core of an Intel Platinum 8628 CPU, and are the average over 5 models on the test set of 100 graphs. The total time includes the entire inference process including both model forward pass and environment step, while the environment time reports only the time spent in the environment step function. Notably, the NAR implementation is written in JAX (Bradbury et al., 2018), which

Table 15: Chosen hyperparameter values for each domain.

| Experiment | Processor | Aggr | Pooling | L | LR (BC) | Batch (BC) | LR (PPO) | Batch (PPO) |
|---|---|---|---|---|---|---|---|---|
| | | | *CLRS* | | | | | |
| BFS | MPNN | Max | Mean | 2 | 1.0e-3 | 16 | - | - |
| DFS | TripletMPNN | Sum | Mean | 5 | 5.0e-4 | 16 | - | - |
| Bellman-Ford | MPNN | Max | Mean | 4 | 5.0e-2 | 128 | - | - |
| MST-Prim | TripletMPNN | Max | Max | 3 | 1.0e-3 | 64 | - | - |
| | | | *TSP* | | | | | |
| BC | MPNN | Max | Max | 4 | 1.0e-3 | 64 | - | - |
| PPO | MPNN | Max | Max | 2 | - | - | 5.0e-4 | 32 |
| WE + PPO | MPNN | Max | Max | 2 | 1.0e-3 | 32 | 5.0e-4 | 32 |
| | | | *MVC* | | | | | |
| BC | MPNN | Sum | Max | 5 | 1.0e-3 | 8 | - | - |
| PPO | MPNN | Sum | Mean | 2 | - | - | 5.0e-4 | 64 |
| | | | *RGC* | | | | | |
| PPO (BA-R) | MPNN | Sum | Mean | 4 | - | - | 5.0e-4 | 128 |
| PPO (ER-R) | MPNN | Sum | Mean | 4 | - | - | 5.0e-4 | 128 |
| WE + PPO (BA-R) | MPNN | Sum | Mean | 3 | 1.0e-3 | 8 | 5.0e-4 | 128 |
| WE + PPO (ER-R) | MPNN | Sum | Max | 3 | 1.0e-4 | 8 | 5.0e-4 | 128 |
| PPO (BA-T) | MPNN | Sum | Max | 2 | - | - | 5.0e-4 | 128 |
| PPO (ER-T) | MPNN | Sum | Mean | 2 | - | - | 5.0e-4 | 32 |
| WE + PPO (BA-T) | MPNN | Sum | Max | 2 | 5.0e-2 | 16 | 5.0e-4 | 128 |
| WE + PPO (ER-T) | MPNN | Sum | Mean | 2 | 5.0e-2 | 8 | 5.0e-4 | 32 |

allows for compilation of the model functions, while GNARL is implemented in PyTorch Geometric (Fey & Lenssen, 2019) and Gymnasium (Towers et al., 2024), so a direct comparison is challenging.

Table 16: Inference time (seconds per graph) for GNARL and NAR models on graphs with $|V| = 64$.

| Method | Inference | Environment | Total |
|---|---|---|---|
| | *BFS* | | |
| NAR | $0.0146_{\pm 0.0007}$ | - | $0.0146_{\pm 0.0007}$ |
| GNARL (all) | $1.43_{\pm 0.03}$ | $0.307_{\pm 0.007}$ | $1.73_{\pm 0.02}$ |
| | *DFS* | | |
| NAR | $0.0234_{\pm 0.0014}$ | - | $0.0234_{\pm 0.0014}$ |
| GNARL (all) | $2.65_{\pm 0.11}$ | $0.199_{\pm 0.015}$ | $2.85_{\pm 0.12}$ |
| | *Bellman-Ford* | | |
| NAR | $0.0149_{\pm 0.0002}$ | - | $0.0149_{\pm 0.0002}$ |
| GNARL (all) | $8.52_{\pm 1.02}$ | $3.97_{\pm 0.72}$ | $12.5_{\pm 1.6}$ |
| GNARL (success only) | $5.56_{\pm 1.38}$ | $3.66_{\pm 0.78}$ | $9.22_{\pm 2.16}$ |
| GNARL (failure only) | $14.8_{\pm 0.2}$ | $4.65_{\pm 0.73}$ | $19.5_{\pm 0.9}$ |
| | *MST-Prim* | | |
| NAR | $0.0161_{\pm 0.0003}$ | - | $0.0161_{\pm 0.0003}$ |
| GNARL (all) | $640_{\pm 248}$ | $14.0_{\pm 4.9}$ | $654_{\pm 253}$ |
| GNARL (success only) | $67.6_{\pm 5.6}$ | $1.76_{\pm 0.27}$ | $69.3_{\pm 5.8}$ |
| GNARL (failure only) | $1220_{\pm 54}$ | $27.0_{\pm 2.7}$ | $1240_{\pm 57}$ |

A large factor influencing inference time is the level of parallelism built into the representation of the algorithm. A more parallel representation leads to shorter episode lengths, reducing both the

number of inference steps and the number of environment steps required during evaluation. This design choice is further discussed in Appendix B.3. Additionally, the inference time of GNARL is influenced by the success rate of the model for certain problems where successful episodes terminate earlier, reducing the average episode length. The number of steps in an unsuccessful episode in MST-Prim is approximately $18\times$ longer than a successful episode, leading to a large increase in inference time due to the low average success rate. This effect is also present in Bellman-Ford, though to a lesser extent due to the lower horizon of the environment and higher success rate of the models.

### F.3 TSP INFERENCE TIME COMPARISON

Table 17 shows the inference time per graph on the test data for the TSP, using GNARL and the MPNN model from Georgiev et al. (2024a). For the Georgiev et al. (2024a) model, we use a beam search width of 1280, as this is the width used to achieve the results reported in Table 3. The results were obtained on a single core of an Intel Platinum 8628 CPU, and are the average over 5 models. For the GNARL model, inference was achieved for sizes up to 200 using 4GB of memory, with 20GB required for size 1000. For the Georgiev et al. (2024a) model, inference was performed using 20GB of memory for sizes up to 200, with 64GB required for size 1000. The Georgiev et al. (2024a) model is primarily implemented using PyTorch Geometric (Fey & Lenssen, 2019), providing a fairer comparison to GNARL than the JAX-based NAR implementation used in Appendix F.2.

The results demonstrate that GNARL is able to achieve much faster inference times, due to the lack of reliance on beam search. The beam search uses approximately half of the total inference time of the NAR approach at each size. In comparison, the environment step of the GNARL model requires only a small fraction of the total inference time, making the overall approach faster despite the similar amount of time required for the model forward pass.

Table 17: Inference time per graph (in seconds) for GNARL and NAR models on TSP test data.

| | | Georgiev et al. (total) | Georgiev et al. (beam search) | GNARL (total) | GNARL (environment) |
|---|---|---|---|---|---|
| **Test size** | 40 | $1.17_{\pm 0.02}$ | $0.553_{\pm 0.012}$ | $0.501_{\pm 0.016}$ | $0.0407_{\pm 0.0051}$ |
| | 60 | $3.95_{\pm 0.14}$ | $1.92_{\pm 0.09}$ | $1.74_{\pm 0.05}$ | $0.0816_{\pm 0.0221}$ |
| | 80 | $9.94_{\pm 0.25}$ | $5.01_{\pm 0.16}$ | $3.97_{\pm 0.08}$ | $0.108_{\pm 0.008}$ |
| | 100 | $18.7_{\pm 0.5}$ | $9.26_{\pm 0.36}$ | $7.62_{\pm 0.18}$ | $0.159_{\pm 0.014}$ |
| | 200 | $149_{\pm 5}$ | $73.4_{\pm 3.4}$ | $71.0_{\pm 1.6}$ | $0.661_{\pm 0.035}$ |
| | 1000 | $12800_{\pm 171}$ | $2140_{\pm 15}$ | $8930_{\pm 150}$ | $68.8_{\pm 1.7}$ |

In combinatorial optimisation domains such as the TSP, it is much more difficult to design a highly parallel representation of the problem in comparison to classical algorithms. As a result, NAR-based methods do not have an advantage in episode length for these problems. As secondary solution resolving methods such as beam search are often required to achieve high-quality solutions, GNARL is able to achieve much faster inference times due to its lack of reliance on such methods.

## G ADDITIONAL EXPERIMENTS

### G.1 GRAPH ACCURACY AND SOLUTION CORRECTNESS FOR NAR MODELS

Given a single node ordering, the graph accuracy of a model is calculated as the proportion of graphs for which all output predictions match the expected solution. Conversely, solution correctness is calculated as the proportion of graphs for which the predicted solution could be produced by the reference algorithm under any node ordering. If an algorithm has a unique solution for each input graph, then graph accuracy and solution correctness are equivalent. However, when multiple solutions are possible for the algorithm, graph accuracy is a lower bound for solution correctness, as it is possible that a model could predict a correct solution which does not strictly correspond to the expected output label.

For Bellman-Ford and MST-Prim, edge weights are randomly sampled from a uniform distribution, meaning the probability of multiple solutions existing is 0, so graph accuracy and solution correct-

ness are equivalent. However, for BFS and DFS, multiple solutions are possible for a given input graph. In BFS, any tree with root $v_s$ and with all nodes at the correct depth from $v_s$ is a valid solution. In DFS, any valid depth-first traversal of the graph is a valid solution, with no restrictions on the root node, meaning many solutions exist.

We evaluate the graph accuracy and solution correctness of models for BFS and DFS produced by the TripletMPNN trained per Ibarz et al. (2022) in Table 18, with the evaluation run over 100 graphs with $|V| = 64$. The results confirm that graph accuracy is not directly equivalent to solution correctness for algorithms with many possible solutions. We also include the value of the estimated graph accuracy based on the node accuracy (micro-$F_1$ score) alone.

Table 18: Graph accuracy and solution correctness for TripletMPNN over 5 seeds.

| | Solution correctness (any ordering) | Graph accuracy (single ordering) | Estimated graph accuracy (micro-$F_1^{|V|}$) |
|---|---|---|---|
| BFS | $100.0_{\pm 0.0}\%$ | $87.4_{\pm 10.1}\%$ | $85.0_{\pm 12.7}\%$ |
| DFS | $24.2_{\pm 28.6}\%$ | $0.0_{\pm 0.0}\%$ | $0.0_{\pm 0.0}\%$ |
| Bellman-Ford | $12.4_{\pm 6.5}\%$ | $12.4_{\pm 6.5}\%$ | $14.2_{\pm 5.3}\%$ |
| MST-Prim | $2.6_{\pm 4.2}\%$ | $2.6_{\pm 4.2}\%$ | $0.6_{\pm 0.9}\%$ |

### G.2 TSP with Weak Expert Warm-Starting

In some circumstances, it is not possible to allocate a large computational budget to PPO training. In such cases, it can be beneficial to pre-train the policy using expert demonstrations, even if only a weak expert is available. We simulate such a scenario for the TSP in which the weak expert (WE) policy greedily selects the next node to maximise the objective function. In Table 19, we compare the results of training GNARL using PPO for $10^6$ steps against GNARL trained using only the weak expert for $10^5$ episodes (WE), and GNARL trained using the weak expert for $10^5$ episodes followed by PPO fine-tuning for $10^6$ steps (WE+PPO). We also include results for GNARL trained using the weak expert for $10^5$ episodes followed by PPO fine-tuning for $10^6$ steps using an augmented validation set (WE+PPO+Val). This validation set contains the original validation set with $|V| = 20$ and adds 10 graphs of each size $|V| \in \{40, 60, 80\}$. The results demonstrate that pre-training using the weak expert policy can improve performance at scales similar to those seen in training, and that explicitly selecting models for OOD performance can further improve performance at larger scales. Interestingly, even the model trained using only the weak expert is able to outperform Georgiev et al. (2024a) at OOD scales above $4\times$, demonstrating the scalability benefits of the GNARL architecture.

Table 19: Percentage above the optimal baseline on the TSP for models trained using demonstrations from a weak (greedy) expert, with further fine-tuning via PPO.

| | Test size | | | | | |
|---|---|---|---|---|---|---|
| **Model** | 40 | 60 | 80 | 100 | 200 | 1000 |
| GNARL$_{\text{PPO}}$ | $8.8_{\pm 0.3}$ | $11.0_{\pm 0.3}$ | $12.3_{\pm 1.5}$ | $14.8_{\pm 0.9}$ | $17.6_{\pm 2.3}$ | $26.1_{\pm 7.2}$ |
| GNARL$_{\text{WE}}$ | $15.3_{\pm 10.5}$ | $17.3_{\pm 8.2}$ | $17.7_{\pm 6.8}$ | $19.5_{\pm 5.7}$ | $21.1_{\pm 6.7}$ | $24.4_{\pm 2.7}$ |
| GNARL$_{\text{WE + PPO}}$ | $8.5_{\pm 0.4}$ | $11.2_{\pm 1.0}$ | $11.9_{\pm 1.0}$ | $13.9_{\pm 0.9}$ | $17.3_{\pm 1.8}$ | $29.3_{\pm 10.1}$ |
| GNARL$_{\text{WE + PPO+Val}}$ | $8.6_{\pm 0.2}$ | $10.4_{\pm 0.3}$ | $11.8_{\pm 0.8}$ | $13.3_{\pm 0.9}$ | $15.9_{\pm 0.6}$ | $20.5_{\pm 3.4}$ |

### G.3 MVC Evaluation Details

For the MVC environment, we use the objective value of the solution generated by an approximation algorithm (Khuller et al., 1994) as the reference by which other models are normalised. The objective value of this algorithm is given by $J_{\text{approx}} = -\sum_{v \in C_{\text{approx}}} w_v$, where $C_{\text{approx}}$ is the cover produced by the approximation algorithm.

In the design of the MVC MDP, the episode terminates as soon as a valid cover is found. Since the approximation algorithm may include extraneous nodes in the cover $C_{\text{approx}}$, it is possible that

a valid cover $C \subset C_{\text{approx}}$ exists. Thus, a model which exactly replicates the covers produced by the approximation algorithm may, by chance, select nodes in such an order that $C$ is found and the MDP terminates before $C_{\text{approx}}$ is selected, achieving a better objective function value.

We evaluate the effect of this truncation at different graph sizes in Table 20. Here, the approximation algorithm is evaluated, with nodes in $C_{\text{approx}}$ chosen with uniform probability, and normalised against the full cover objective $J_{\text{approx}}$. The overall effect is small.

Table 20: Objective ratio for MVC under the expert policy given by the approximation algorithm.

|  | **Test size** | | | | | | |
|---|---|---|---|---|---|---|---|
|  | 16 | 32 | 64 | 128 | 256 | 512 | 1024 |
| $J/J_{\text{approx}}$ | 0.9968 | 0.9938 | 0.9976 | 0.9991 | 0.9994 | 0.9996 | 0.9999 |

