# OpenReview forum: "Tackling GNARLy Problems: Graph Neural Algorithmic Reasoning Reimagined through Reinforcement Learning"
_ICLR.cc/2026/Conference — Submitted to ICLR 2026_

### Official Review · Reviewer_dAeN · 2025-10-16

**Soundness:** 3
**Presentation:** 2
**Contribution:** 3
**Rating:** 4
**Confidence:** 3

**Summary:**

This paper reframes the NAR as an RL process and builds an architecture that is applicable to problems with multiple correct answers, NP-hard problems, and problems without expert algorithms. Performances are evaluated on several CLRS-30 problems and the robust graph construction task.

**Strengths:**

Redefining the algorithm learning process as a trajectory optimization problem is a promising paradigm. Experiments show that the proposed method solves key problems of NAR methods.

**Weaknesses:**

1. The introduction of the proposed method, model variants, and experimental metrics is not clear enough. Sometimes it's hard to understand the content due to a loss of details.

2. The time consumption compared with other methods is lacking.

3. The performance of the proposed method is still inferior compared to advanced Non-NAR methods.

**Questions:**

1. For problems without expert algorithms, how do you compute the rewards?

2. In your framework, how do you train GNARL-BC and GNARL-PPO in detail? I think it should be presented in a concentrated manner rather than scattered across different sections.

3. Line 316 says "meaning that not all CLRS-30 graph problems are representable for the time being". Can you show some examples in CLRS-30 that GNARL can not handle, and is this a limitation of GNARL?

4. How much computational overhead is brought by training the critic model? Can you compare the training time consumption of GNARL and other NAR methods?

5. What is the meaning of the metric "TSP percentage above optimal objective"? Is it the lower the better?

6. In the Limitation Section, it says "GNARL relies on the environment during execution, creating a performance bottleneck. ". Can you explain this in detail?

7. On evaluation of CLRS-30, it says "Rodionov & Prokhorenkova (2025) report 100% graph accuracy on the BFS, DFS, and MST-Prim problems. ". Why is this baseline removed from the comparison in Table 2?

8. Why is GNALR-PPO not considered in Table 2?

9. In Table 3, why do you use only 10% of the training data? What is the performance comparison when using the full training data?

---

> ### Author Response · Authors · 2025-11-21
>
> We thank the reviewer for the careful revision of our manuscript. We have responded to each comment below and incorporated changes from your valuable suggestions.
>
> - Weaknesses
>
> 1. We apologise for the loss of detail. We have made changes to the manuscript to improve clarity and add information. We appreciate your insightful questions which have allowed us to address many of the shortcomings.
>
> 2. We have now added experiments regarding the inference time in Appendices F.2 and F.3. We demonstrate that GNARL is advantageous in inference time compared to NAR-based combinatorial optimisation methods, due to the lack of reliance on a secondary search method like beam search. At the moment, we perform worse in inference time on the CLRS-30 Benchmark problems due to the serial nature of GNARL. However, we note that GNARL is able to achieve a much higher success rate, so this is an acceptable trade-off.
>
> 3. We agree that both GNARL and NAR are presently unable to compete with advanced problem-specific methods. The advantage in a NAR or GNARL-based approach is the formulaic construction of the problem, making any algorithm or combinatorial optimisation problem easily approachable where no prior problem-specific method exists. This means that for new problems, using the GNARL framework a reasonable solution can be found quickly without having to acquire large amounts of domain knowledge.

---

> > ### Author Response · Authors · 2025-11-21
> >
> > - Questions
> >
> >     1. Problems without an expert algorithm are typically defined by an objective function which is to be optimised, for example the RGC problem. In order to compute the reward at each step, we use the difference in the objective function evaluated before and after the action, as described in line 239 (previously 231). If the objective function can only be evaluated in a final state, we use this as the (sparse) reward.
> >
> >     2. We review BC and PPO in detail in Section 3.2. Figure 1B illustrates the difference between the two ways of training GNARL. Other RL and IL algorithms are also compatible with GNARL, and algorithmic advances would also mean better performance for our framework. We have updated Section 4.3 to clarify the training method with reference to specific equations and welcome further specific suggestions to improve clarity in this section. We appreciate the suggestion to unify the description of the training method, however we feel that Section 3.2 is general RL methodology rather than a contribution of GNARL, so the separation avoids confusing the reader.
> >
> >     3. In the implementation of our GNN layers, we use a sparse graph representation which allows us to scale to large graphs (thousands of nodes). By using the sparse representation, we require that all edge features are assigned to edges which actually exist in the graph. A CLRS-30 problem which breaks this convention is MST-Kruskal, in which one of the features is a pointer between a node and its root in the current spanning tree. It is not guaranteed that an edge exists in the base graph between a node and its root in the tree, so this feature cannot be straightforwardly represented in a sparse message-passing architecture. This trade-off was introduced by \[1] and has been used in several works since \[2]. Importantly, the trade-off is simply due to the implementation of the GNN and is not inherent to GNARL. We could use a message passing architecture which considers the complete graph, but this scales very poorly. We have updated the corresponding text to highlight this.
> >
> >     4. We have added an experiment in Appendix E.3 to evaluate the overhead of training the critic. The results demonstrate that training the critic module introduces a 36% overhead in training time. In the main paper, the only experiments that train the critic module when using BC are the WE+PPO experiments in the RGC domain. In the remaining experiments, no fine-tuning was used so the critic network was disabled. We also evaluate the training time of GNARL vs NAR methods in Appendix E.2. Here we find that the training time of GNARL is heavily dependent on the MDP definition and the number of steps taken in the problem. Note that due to differences in implementation frameworks it is difficult to perform a direct comparison.
> >
> >     5. Yes, the optimal objective is the path length of the tour found by the Concorde solver, which we use as a baseline to compare the longer paths found by sub-optimal methods. We have rephrased the caption to make this clearer and added arrows to indicate the direction of “better” for each metric.
> >
> >     6. During execution, a state is given as input to the policy network, which outputs an action distribution. The resultant action is applied in the environment, which produces the next state. This means that execution depends both on the GNN implementation and also the environment, which is here implemented as a Gymnasium vectorised environment. In contrast, at inference time, NAR can be entirely executed by repeated application of the policy network and does not rely on an external function to produce the next state. We have added experiments in Appendices F.2 and F.3 which demonstrate the impact of the environment on total inference time. These experiments reveal that the environment contributes to between 5 and 40% of the total evaluation time. The inference time experiments in the CLRS problems reveal that a major difference in inference time between NAR and GNARL is due to the lack of parallelism in the GNARL formulation. However, in combinatorial optimisation domains, this advantage is not present, as such domains are much harder to parallelise, and we see here that GNARL is more efficient due to its lack of reliance on a secondary resolution method like beam search.

---

> > > ### Author Response · Authors · 2025-11-21
> > >
> > > 7. Rodionov & Prokhorenkova (2025) present a technique which is highly optimised for achieving neural execution of classical algorithms, but the methods they apply rule out extensibility to combinatorial optimisation-style problems, or predicting distributions. Table 2 aims to compare methods which could in theory be adapted for at least one of these two goals. We include the reference for completeness and to demonstrate that perfect performance is possible in practice.
> > >
> > > 8. Table 2 presents the performance of the approach for learning classical algorithms. The classical algorithms do not have a defined reward function, so RL cannot be applied. While some algorithms can be assigned an objective function (e.g. sum of shortest paths found so far for Bellman-Ford), training using RL in these instances would not necessarily align with the original algorithm. This approach may be of interest for algorithmic discovery, but we reserve this for future work and here focus on the goal of learning to imitate a reference algorithm.
> > >
> > > 9. We use a subset of the training data for both storage and training time efficiency. We found that 10% of the training data was sufficient and representative of the full dataset, as early tests demonstrated no significant difference in performance using the smaller set. We run our experiments to mainly compare the performance of our model against different baselines, and thus, we don't expect any meaningful changes in the _relative_ performance between them when scaling to the full training set.
> > >
> > > \[1] Julian Minder, Florian Grotschla, Joel Mathys, and Roger Wattenhofer. SALSA-CLRS: A sparse and scalable benchmark for algorithmic reasoning. In LoG, 2023.
> > >
> > > \[2] Dobrik Georgiev, Joseph Wilson, Davide Buffelli, and Pietro Lio. Deep Equilibrium Algorithmic Reasoning. In NeurIPS, 2024.

---

### Official Review · Reviewer_svJy · 2025-10-27

**Soundness:** 3
**Presentation:** 2
**Contribution:** 3
**Rating:** 6
**Confidence:** 3

**Summary:**

In this work the authors propose the Graph Neural Algorithm Reasoning with RL (GNARL) framework. This approach casts algorithmic processes as trajectories in MDPs. By formulating the problem as an MDP, they leverage RL techniques to improved performance in replicating performance on algorithmic reasoning tasks. In contrast to other NAR approaches, the authors use MDPs to ensure valid solutions, cast P and NP graph problems into the same formalism, and learn examples without an expert algorithm.

**Strengths:**

The core idea of this paper is good, and appears quite promising.

The description of the need the authors hope to fill is clear and well-motivated.

The paper’s results are very even-handed, with a clear assessment of the limitations of the current work.

**Weaknesses:**

If my understanding is correct, the key insight is the formulation of the problem as an MDP. Thus, section 1 is a critical section of the paper. This section was difficult to parse. It seems that the transition function is algorithm-dependent, but many of the other elements are not. Perhaps the authors could provide a compact definition of a GNARL MDP that is similar to the MDP tuple definition found at the beginning of 3.2. Then the GNARL MDP tuple could be defined in terms of graph elements, with notes about which elements are the same for all problems, and which vary according to problems.

In general, it strikes me as difficult to think about how to formulate an algorithmic problem as an MDP, even after reading this paper. It seems to me that a more general framing of the approach would be helpful. Or perhaps this difficulty is the core trade-off, as indicated in the limitations section?

For me, the second and third paragraphs of 3.2 felt superfluous, except perhaps the definitions of the acronyms BC and IL. If the authors need more room to describe their method, I think this section could be trimmed substantially.

**Questions:**

See weaknesses.

---

> ### Author Response · Authors · 2025-11-21
>
> Thank you for your thoughtful review and constructive feedback. We address each point below and have revised the manuscript.
>
> - In the GNARL MDP, the state features and independent parts of the transition function common to the GNARL framework are the phase $p$ and the previously selected node $\psi\_v$. All other state features and their transitions are determined by the problem. We use the state features as they are defined in the CLRS-30 Benchmark. For example, the phase feature $p$ and its evolution during execution is defined by the GNARL architecture, whereas a feature like $\texttt{pred}$ is defined as a hint for DFS in NAR, and its evolution in the transition function is inspired by the original DFS algorithm. Some elements, such as $\texttt{adj}$ and $v\_s$ are common in several algorithms, but these are part of the algorithm specification and not required by GNARL. We have updated the paragraph in what is now line 232 to make this distinction clearer.
>
> - You are correct that the difficulty in precisely defining an MDP is the core trade-off of the GNARL approach. While it is typically straightforward to extract a transition function when given the algorithm and state variables, there is still a core element of human design. We have discussed this in greater depth in the new Appendix B.
>
> - Thank you for your suggestion. At present, it is our understanding that many readers who are familiar with the NAR literature may have limited knowledge or familiarity with reinforcement learning, as NAR is a supervised learning approach. We have included this section to give sufficient background and make the paper accessible to these readers. Further insights on our methodology, experiments, and results are included in the Appendix.

---

### Official Review · Reviewer_vU6f · 2025-10-27

**Soundness:** 3
**Presentation:** 2
**Contribution:** 3
**Rating:** 6
**Confidence:** 4

**Summary:**

Neural Algorithmic Reasoning (NAR) involves training neural networks to learn to exectue algorithms. This is typically done by executing the algorithm and collecting  supervision signal (e.g., intermediate variable values) at every step of the algorithm. This data can then be used to train the neural network by supervising the output at every step. This paper proposes a new approach which views algorithms as Markov Decision Processes with a state, a set of actions, and a transition function. This then allows to train NAR models to produce the right action at every step, hence moving the supervision signal from the domain of outputs to those of actions. Training then happens though reinforcement learning methods. In particular the authors propose to use either imitation learning (when an expert policy is available) or proximal policy optimization.
The proposed method is evaluated on standard algorithms for NAR from the CLRS dataset, a combinatorial optimization algorithm (travelling salesman), and an NP algorithm (minimum vertex cover). The baselines are provided by popular NAR methods for CLRS, and specialized approaches for the other scenarios.

**Strengths:**

- Viewing NAR as an MDP is a very intuitive yet novel idea
- The method can in principle be applied to several graph algorithms
- The experimental results show important improvements over previous methods

**Weaknesses:**

- There are some existing methods which train NAR models without supervising at every step that have not been considered (e.g., "Deep Equilibrium Algorithmic Reasoning", Georgiev et al., 2024l "Deep equilibrium models for
algorithmic reasoning", Xhonneux et al. 2024). These works should be cited at the very least
- I found some parts of the text a bit unclear (see questions below)
- defining the MDP for a given algorithm is not always trivial and there may be more possible MDPs for any given algorithm

**Questions:**

- Are there ways to extend the method to algorithms in which the Markov property doesn't hold?
- Could you explain how the proto action works? I found it quite unclear from the current text
- Does the translation from "encode-process-decode" to "encode-process-act" lead to a much higher number of "steps"? From what is shown in Appendix B it seems that this is the case. Can this be an issue? I feel like there should be a discussion about this aspect in the paper
- Could the authors please expand on the statement that the proposed method "can handle multiple correct solutions"? Is this because instead of having an end-to-end "gold" trajectory, one can just apply 1 step from the current state and then use it for supervision?
- Reinforcement learning is notoriously challenging to apply in practice. Could you please add plots on the training stability of the proposed method and some discussion on the number of training samples required with respect to supervised training?
- Could the authors please expand on what is meant with "we estimate the graph accuracy as micro-F1|"? This last paragraph in Page 6 is quite unclear to me
- Is there an "automatic" procedure that can be used to define an MDP from a given algorithm?

**Details Of Ethics Concerns:**

no concern

---

> ### Author Response · Authors · 2025-11-21
>
> Thank you for taking the time to review our manuscript and for your insightful feedback. We have addressed each comment below and further revised our manuscript.
>
> - Thank you for the suggested references. These provide useful contextualisation of other indirectly supervised approaches in NAR. We have added these to the manuscript.
>
> - Regarding the design of an MDP for an algorithm, we have added a new Appendix B which describes the process in more detail with a worked example, and also discusses the design choices encountered where there may be multiple possible MDP representations. We agree that more detail on this facet should have been included in the original manuscript.
>
> - Questions:
>
> 1. In a classical algorithm, the execution output of a given step depends on the present internal state of the algorithm. This aligns with the Markov property, even if the algorithm is non-deterministic. We can construct two scenarios in which the algorithm’s steps may not be Markovian: 1) The state is not fully observable; 2) Some required computations in a single step are beyond the representation ability of the network. In order to learn these algorithms using GNARL, we could improve the problem representation by adding more features, for example storing intermediate values which would help with complex computations. Alternatively, the method could be extended by adding recurrent features, such as those present in the original NAR formulation. At this stage, we have not discovered any algorithms which necessitate recurrency in the architecture. We have discussed some design considerations around this in Appendix B.2.
>
> 2. We have added a diagram in Appendix A.1. Please let us know if this is still unclear and we will seek to further elaborate.
>
> 3. Your intuition is correct - the NAR framing allows multiple operations to occur in a single step, as labels for all nodes form the output of the model, whereas in GNARL these operations must occur in series. We have added a discussion of this in the new Appendix B.3.
>
> 4. In a NAR trajectory, each input corresponds to exactly one correct solution, which is the solution produced by the reference algorithm given a particular node ordering, encoded by the \`pos\` feature. The output features of NAR are only able to represent a single value rather than a distribution. In order to attempt to produce a different solution using NAR, one might try to use the same input features but a different permutation of \`pos\`. However, \[1] demonstrates that this does not work in practice. In contrast, GNARL predicts a \_distribution\_ over actions, rather than predicting a single action at each step (analogous to vanilla NAR). This means that, by varying the sampling temperature, multiple solutions can be straightforwardly produced. We have relocated Appendix E.2, which examines this for BFS and DFS, into the main text, as this is an important feature of the GNARL framework that we wish to highlight.
>
> 5. In practice, we found that PPO was quite stable. We have added a comparison of the training performance for PPO and BC on the TSP to Appendix E.1. The results demonstrate that PPO converges faster in terms of training time, but this does not necessarily translate to generalisation performance.
>
> 6. In most NAR works, metrics for success are measured using node accuracy, also called micro-F1. This metric measures the proportion of nodes for which the label was correctly predicted. We consider graph accuracy as a metric, which corresponds to the proportion of graphs for which _all_ nodes had a correct label prediction (including alternative solutions). As the source code for the cited works was not available, we were unable to directly measure the graph accuracy of their methods, so we chose to estimate it. Assuming each node’s probability of being correctly predicted is independent, we can approximate graph accuracy by raising the node accuracy to the number of nodes in the graph, hence we arrive at micro-F1$^{|V|}$. We further discuss graph accuracy and the accuracy of this approximation in Appendix G.1.
>
> 7. All of the MDPs in this paper were defined manually by comparing to a reference implementation of the algorithm being learned. We have elaborated on this process in Appendix B. In future work, we intend to investigate methods which could be used to learn the MDP representation automatically.
>
> \[1] Zeno Kujawa, John Poole, Dobrik Georgiev, Danilo Numeroso, Henry Fleischmann, and Pietro Lio.\` Neural algorithmic reasoning with multiple correct solutions. In KDD Workshop on Machine Learning on Graphs in the Era of Generative Artificial Intelligence, 2025.

---

> > ### Comment · Reviewer_vU6f · 2025-11-24
> > **Response to rebuttal**
> >
> > I thank the authors for answering all my questions and for adding new appendices to explain some of the concepts. I think the newly added sections are very clear. I am happy to keep my positive score and I will increase my rating for the presentation.

---

### Official Review · Reviewer_qSb5 · 2025-10-31

**Soundness:** 1
**Presentation:** 2
**Contribution:** 1
**Rating:** 2
**Confidence:** 2

**Summary:**

The paper presents a rephrasing of the reinforcement learning or Markov Decision Process in the language of Neural Algorithmic Reasoning (NAR).

The authors present the general mapping and experiments on the CLRS-30 benchmark on Depth-first or Breadth-First Searches, on TSP instances, on Minimum Vertex Cover (MVC), and robust graph construction (RGC).

The model can be trained similarly to RL, using Imitation training or using proper RL methods (e.g.PPO)

The paper's contributions are:
1. to show the mapping between NAR and MDP
2. to evaluate and compare the performance on 4 learning tasks

**Strengths:**

The paper is well written, the related works are mostly covered, and a relatively good number of evaluations.

**Weaknesses:**

There is no reference in the related work at least to the GFlowNet approach.

The approach is meant to be NAR, but the mapping depends on the problem: "A(s) are specified for each problem", "the horizon $h$ is defined by the problem".

In general, it is hard for me to really understand the difference with respect to using RL. The "framework" is to define the State space $\mathcal S=\mathcal T \times \mathcal F$, with $\mathcal T$ the imposta space and $\mathcal F$ the state space of the graph.

I probably also have a problem understanding what is different in the general NAR framework, which seems to be the idea of having an encoder, a decoder, and some iteration in the latent space.

While I find CO really fascinating, I am not sure I really appreciated the NAR perspective; therefore, it is hard for me to evaluate this work.

**Questions:**

Based on my previous analysis, the author shall therefore clarify better 1) what the actual contribution is compared to RL, 2), explain the difference to RL, and 3) also explain the difference with GFlowNet, which also defines an MDP over a graph of states.

**Details Of Ethics Concerns:**

No concern

---

> ### Author Response · Authors · 2025-11-21
>
> We thank the reviewer for the valuable feedback and questions.
> - We reiterate here that:
>     - In the NAR blueprint, algorithms are learned via supervised learning using sets of labels from an expert algorithm. NAR provides structure to this supervised problem by breaking down algorithmic trajectories into a series of steps, with different “hints” at each step reflecting the internal state of the algorithm. However, **NAR does not make use of an MDP** or any other structured decision making model. GNARL (our novel method) casts the algorithm as an MDP, allowing RL methods to be applied to learning the algorithm. GNARL builds on of the techniques of NAR, such as the input/hint space definitions and the encoder and processor, allowing us to easily transform existing NAR problems into something that can be tackled using GNARL.
> - Contribution and difference to RL:
>     - It is important to note that GNARL is not a new Reinforcement Learning algorithm (like PPO or DQN), but rather a novel frameworking (combining the algorithm MDP formulation and a related neural architecture), designed to bridge the gap between the Neural Algorithmic Reasoning framework and RL, as described in Section 1. Using GNARL, it is possible to apply existing MDP solution techniques, such as RL and IL, to learning algorithms. One of the main contributions of the paper is to demonstrate how to take an algorithm or a CO problem and turn it into an MDP that can be solved with high quality. RL provides the optimisation process, while GNARL provides the interface and inductive biases necessary to apply that process to algorithmic reasoning.
> - GFlowNet main differences:
>     - GFlowNet is a tool for generating a diverse set of samples which score highly on an objective function, for example, generating candidate molecules with high binding affinity to a protein. The main focus of GFlowNet is the diversity of generated samples, with reward as a driving but secondary factor. When solving a CO problem with GNARL, the primary focus is the maximisation of the objective function, with generating alternative solutions being a secondary objective. However, as previously mentioned, GNARL is a design framework rather than a solution method, and it would be possible to apply GFlowNets instead of PPO to solve a problem framed using GNARL, provided it meets the restrictions of GFlowNets, namely:
>         - GFlowNets require the state transitions to form a DAG (previous states cannot be revisited). This can be enforced by adding a timestamp to the states in the GNARL MDP.
>         - GFlowNets require strictly non-negative rewards. Minimisation problems such as the TSP can be approached by transforming the reward, e.g. taking the inverse of the cost.
>     - As the primary purpose of GFlowNets is the construction of diverse solutions, it may be poorly suited to learning algorithmic trajectories where only one solution exists, for example Bellman-Ford (assuming edges have unique weights) and MST. It is likely better suited for CO problems where several solutions might have very similar objective values, or where there is some uncertainty in the true value of the objective function, such as the RGC environment.

---

### Author Response · Authors · 2025-11-12
**Code repository**

Dear reviewers,

Please find the code for this paper here: https://anonymous.4open.science/r/GNARL-D687/

Thank you for your detailed feedback. We will take some time to review the comments and respond.

---

### Meta-Review · Area_Chair_2j4v · 2026-01-02

**Summary:**

The paper introduces GNARL, a framework that reframes Neural Algorithmic Reasoning as a Markov Decision Process, enabling reinforcement and imitation learning methods to be applied. Reviewers found the idea novel and promising, with potential to broaden the applicability of NAR. However, concerns remain.

First, it emerged during the rebuttal that GNARL is 100–10,000× slower than NAR, raising doubts about its practicality for real-world workloads. In addition, GNARL performs worse than problem-specific approaches. Authors admitted this trade-off but did not provide new evidence to close the gap. Given that problem-specific approaches are better, and NAR is dramatically faster, the overall impact is limited.

Second, all four reviewers found the paper difficult to parse with important details omitted. Concerns included: lack of detail in methodology, scattered explanations of training procedures, unclear metrics, and insufficient discussion of limitations. While the rebuttal added appendices and clarified some points, it remains uncertain whether the revisions fully addressed the presentation issues.

Given the combination of dramatically reduced efficiency and persistent presentation issues, I recommend rejection.

**Reviewer Concerns:**

The key outstanding concerns that remain unresolved despite the rebuttal are summarized in the "Summary" field. Other concerns that were addressed during the rebuttal are as follows:

* Handling multiple correct solutions: Reviewers asked how GNARL could produce diverse solutions. The authors clarified that GNARL predicts distributions over actions, enabling multiple solutions via sampling.

* Training stability: Concerns about RL instability were studied for TSP in Appendix E.1, showing PPO’s stability.

* Definition of metrics: The rebuttal clarified the meaning of “graph accuracy” and explained the approximation method for $micro-F1^{|V|}$.

* Missing references: Reviewers noted omissions (e.g., Deep Equilibrium Algorithmic Reasoning, GFlowNets). The authors added these citations and contextualized them in the revised manuscript.

* Evaluation gaps: Key baselines (e.g., Rodionov & Prokhorenkova 2025 with 100% graph accuracy) were omitted from direct comparison in Table 2. The rebuttal acknowledged this but did not integrate them into results.

* Performance vs. non-NAR methods: GNARL still performs worse than advanced problem-specific approaches. Authors admitted this trade-off but did not provide new evidence to close the gap.

**Reviewer Scores:**

Reviewer qSb5: This reviewer initially gave a rating of 2, but with low confidence. Given the uncertainty and limited engagement, it is difficult to predict how the score might have shifted following the rebuttal. My decision therefore relies primarily on the other three reviews.

Reviewer vU6f: The discussion concluded with this reviewer maintaining a rating of 6, indicating that the rebuttal did not alter their overall assessment.

Reviewer svJy: This reviewer did not raise significant concerns and originally rated the paper a 6. Based on the rebuttal, I expect the score to remain unchanged at 6.

Reviewer dAeN: This reviewer initially rated the paper a 4 and requested additional experiments to evaluate efficiency. The rebuttal revealed that the proposed technique is 100–10,000 times slower than NAR, a critical limitation. Placing myself in the reviewer’s position, I do not see the rating improving in light of this finding.

---

### Decision · Program_Chairs · 2026-01-26

Reject